# TemMed-Bench: Evaluating Temporal Medical Image Reasoning in Vision-Language Models

## Abstract

Existing medical reasoning benchmarks for vision-language models primarily focus on analyzing a patient's condition based on an image from a *single* visit. However, this setting deviates significantly from real-world clinical practice, where doctors typically refer to a patient's historical conditions to provide a comprehensive assessment by tracking their changes over time. In this paper, we introduce TemMed-Bench, a multi-task benchmark designed for analyzing changes in patients' conditions between different clinical visits, which challenges large vision-language models (LVLMs) to reason over **tem**poral **med**ical images. TemMed-Bench consists of a test set comprising three tasks – visual question-answering (VQA), report generation, and image-pair selection – and a supplementary knowledge corpus of over 17,000 instances. With TemMed-Bench, we conduct an evaluation of twelve LVLMs, comprising six proprietary and six open-source models. Our results show that most LVLMs lack the ability to analyze patients' condition changes over temporal medical images, and a large proportion perform only at a random-guessing level in the closed-book setting. In contrast, GPT o3, o4-mini and Claude 3.5 Sonnet demonstrate comparatively decent performance, though they have yet to reach the desired level. To enhance the tracking of condition changes, we explore augmenting the input with both retrieved visual and textual modalities in the medical domain. We also show that multi-modal retrieval augmentation yields notably higher performance gains than no retrieval and textual retrieval alone across most models on our benchmark, with the VQA task showing an average improvement of 2.59%. Overall, we compose a benchmark grounded on real-world clinical practice, and it reveals LVLMs' limitations in temporal medical image reasoning, as well as highlighting the use of multi-modal retrieval augmentation as a potentially promising direction worth exploring to address this challenge.

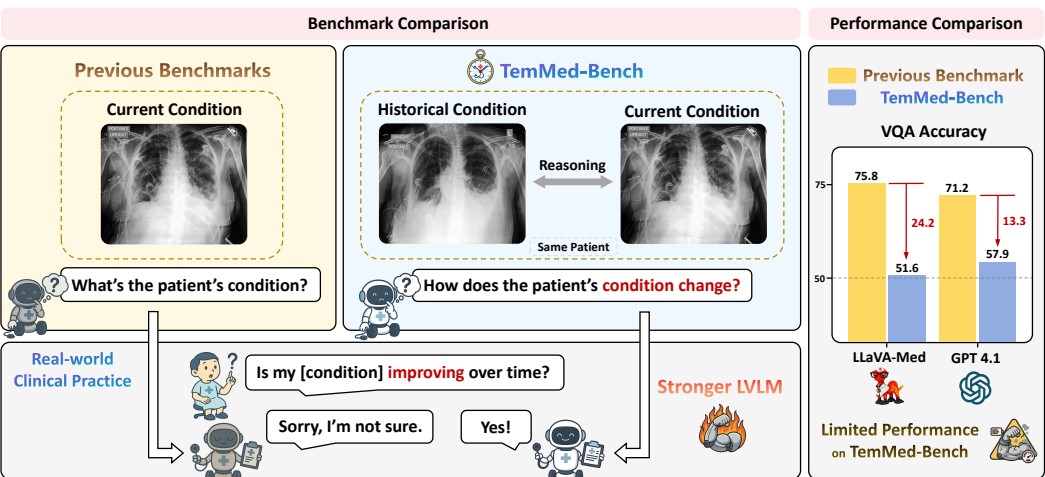

Figure 1: An example from TemMed-Bench. Previous benchmarks (Johnson et al., 2019; Hu et al., 2024) mainly focused on analyzing a single-visit image. However, real-world clinical practice requires doctors to monitor changes in patients' conditions over time. The previous benchmark in the rightmost chart is a VQA variant (Xia et al., 2025) of the MIMIC-CXR dataset (Johnson et al., 2019).

| Benchmarks | Task | Historical Conditions | Multi-Image Input |
|---|---|---|---|
| VQA-RAD (Lau et al., 2018) | VQA | ✗ | ✗ |
| SLAKE (Liu et al., 2021) | VQA | ✗ | ✗ |
| PathVQA (He et al., 2020) | VQA | ✗ | ✗ |
| PMC-VQA (Zhang et al., 2024) | VQA | ✗ | ✗ |
| PubMedVision (Chen et al., 2024) | VQA | ✗ | ✗ |
| OmniMedVQA (Hu et al., 2024) | VQA | ✗ | ✗ |
| Harvard-FairVLMed (Luo et al., 2024) | Report | ✗ | ✗ |
| IU-Xray (Demner-Fushman et al., 2015) | Report | ✗ | ✗ |
| CheXpert Plus (Chambon et al., 2024) | Report | ✗ | ✗ |
| MIMIC-CXR (Johnson et al., 2019) | VQA + Report | ✗ | ✗ |
| TEMMED-BENCH (Ours) | VQA+Report+Image-pair | ✓ | ✓ |

Table 1: Comparison with previous works. TEMMED-BENCH focuses on evaluating LVLMs in temporal reasoning over multiple medical images. VQA: visual question answering; Report: report generation; Image-pair: image-pair selection.

# 1 INTRODUCTION

With the recent developments in Large Vision-Language Models (LVLMs), Medical LVLMs (Med-LVLMs) have shown promise for diagnostic tasks such as disease detection, therapeutic planning, and clinical guidance (Li et al., 2023; Chen et al., 2024; Lin et al., 2025). When evaluating Med-LVLMs, prior benchmarks have suffered from limited modality diversity (Lau et al., 2018; Liu et al., 2021; He et al., 2020), small scale (Lau et al., 2018; Liu et al., 2021; He et al., 2020; Demner-Fushman et al., 2015), and restricted task formats (Zhang et al., 2024; Luo et al., 2024). Although some efforts have mitigated these issues (Hu et al., 2024; Johnson et al., 2019; Chambon et al., 2024), these benchmarks still share a common limitation: they analyze a patient's condition based on a single-visit image. We argue that this limitation prevents the evaluation of Med-LVLMs from capturing real-world clinical practice, where doctors rely on patients' medical histories to comprehensively assess current conditions and track changes over time, as illustrated in Figure 1. This real-world scenario challenges Med-LVLMs to possess strong reasoning abilities over temporal medical images. Regarding this point, some of the most recent works have begun to address it (Yu et al., 2025; Mu et al., 2025; Yang et al., 2025), but they either adopt data sources that have not undergone reliable verification (Yu et al., 2025) or suffer from limitations in the comprehensiveness of their task design and evaluation (Mu et al., 2025; Yang et al., 2025).

We introduce TEMMED-BENCH, the first multi-task benchmark that focuses on comprehensively evaluating the ability of LVLMs to perform temporal reasoning on medical images in both closed-book and open-book settings. Specifically, each sample in TEMMED-BENCH contains images from two different clinical visits of the same patient, requiring the model to analyze the changes in the patient's condition over time. As summarized in Table 1, TEMMED-BENCH features three primary highlights. (1) *Temporal reasoning focus*: Each sample in TEMMED-BENCH includes historical condition information, which challenges models to analyze changes in patient conditions over time. (2) *Multi-image input*: Each sample in TEMMED-BENCH contains multiple images from different visits as input, emphasizing the need for models to process and reason over multiple images. (3) *Diverse task suite*: TEMMED-BENCH comprises three tasks, including VQA, report generation, and image-pair selection. Additionally, TEMMED-BENCH includes a knowledge corpus with more than 17,000 instances to support retrieval-augmented generation (RAG), each comprising two images and one corresponding report of condition changes.

We conducted extensive experiments on TEMMED-BENCH to evaluate six proprietary and six open-source LVLMs. In addition to closed-book evaluation, we also benchmark whether and how LVLMs benefit from retrieval augmentation on TEMMED-BENCH. Beyond augmenting the input with retrieved textual information (Tao et al., 2024; Kumar & Marttinen, 2025; Sun et al., 2025; Xia et al., 2024; 2025), we further explore augmenting the input with both retrieved visual and textual modalities in the medical domain, which remains unexplored.

For the closed-book evaluation, experimental results show that most LVLMs lack the ability to analyze changes in patients' conditions across temporal medical images. In the VQA task, GPT-4o-mini and Claude 3.5 Sonnet achieved accuracies of 79.15% and 69.90%, respectively, while most LVLMs scored below 60%. For the more challenging tasks of report generation and image-pair selection, all LVLMs underperformed, with the highest average BLEU, ROUGE-L, and METEOR score at 20.67 for report generation and a top accuracy of 39.33% for image-pair selection in a three-option setting. These results reveal a fundamental gap in current LVLM training, i.e., lack of focus on temporal image reasoning.

For the retrieval augmentation evaluation, experimental results demonstrate that augmenting input with both visual and textual information substantially boosts performance for most models compared to text-only augmentation. Notably, HealthGPT (Lin et al., 2025) exhibits an accuracy improvement of over 10% in the VQA task when augmented with multi-modal retrieved information. These results confirm that multi-modal retrieval augmentation provides more relevant medical information by retrieving images with similar conditions, highlighting its potential for input augmentation in the medical domain. In addition, we found that while previous benchmarks emphasize pattern recognition and matching for a single-visit image, which can be easily hacked by directly taking top-1 retrieved result as the answer, TEMMED-BENCH is more robust to it due to its reasoning attribute. Further discussion can be found in Appendix C.1.

The main contributions of this paper are as follows: (1) We propose TEMMED-BENCH, the first multi-task benchmark that focuses on comprehensively evaluating the temporal reasoning ability of LVLMs in both closed-book and open-book settings. (2) Comprehensive evaluation of mainstream LVLMs on the three tasks in our benchmark reveals their limitations in temporal reasoning over medical images. (3) We explore multi-modal RAG for the medical domain, and highlight the effectiveness of retrieving multi-modal information to boost the performance of Med-LVLMs.

## 2 TEMMED-BENCH

### 2.1 BENCHMARK OVERVIEW

The key statistics of TEMMED-BENCH are shown in Table 2. TEMMED-BENCH consists of a test set and a knowledge corpus. The test set comprises three tasks: visual question answering (VQA), report generation, and image-pair selection. The formulations of these tasks are as follows:

| Statistic (#) | Questions | Images | Choice |
|---|---|---|---|
| **VQA** | 2,000 | 2 | 2 |
| **Report** | 1,000 | 2 | - |
| **Image-pair** | 862 | 6 | 3 |
| **Corpus** | 17,144 | 2 | - |

Table 2: Key statistics of TEMMED-BENCH. VQA: visual question answering; Report: report generation; Image-pair: image-pair selection. Corpus: knowledge corpus.

*VQA*: An LVLM $\mathcal{M}$ takes a historical image $i_h$, a current image $i_c$, and a textual question $q$ describing the condition change as input, and is required to output a binary answer of "yes" or "no".

*Report Generation*: $\mathcal{M}$ takes $i_h$, $i_c$, and a textual task instruction $q_{inst}$ as input, and is required to output a report that analyzes the changes in condition between these two images.

*Image-pair Selection*: $\mathcal{M}$ is given three image pairs – $I_A$ ([$i_{h1}$, $i_{c1}$]), $I_B$ ([$i_{h2}$, $i_{c2}$]), and $I_C$ ([$i_{h3}$, $i_{c3}$]) – along with a textual question $q$, and is required to output the choice of A, B, or C that best matches the medical statement in $q$.

Besides, each instance in the knowledge corpus follows the format of an image pair ($i_h$, $i_c$) with its corresponding condition change report $t$.

### 2.2 KEY OBSERVATION & RAW DATA COLLECTION

TEMMED-BENCH is built upon a key observation regarding existing medical report generation datasets: while these datasets typically assign each report to a single visit, some reports actually include sentences that describe changes in a patient's condition, rather than just the condition at that visit. More specifically, consider the following examples. If a sentence contains phrases such as "*mild* atelectasis", "*moderate* atelectasis", or "*severe* atelectasis", where the adjectives quantitatively

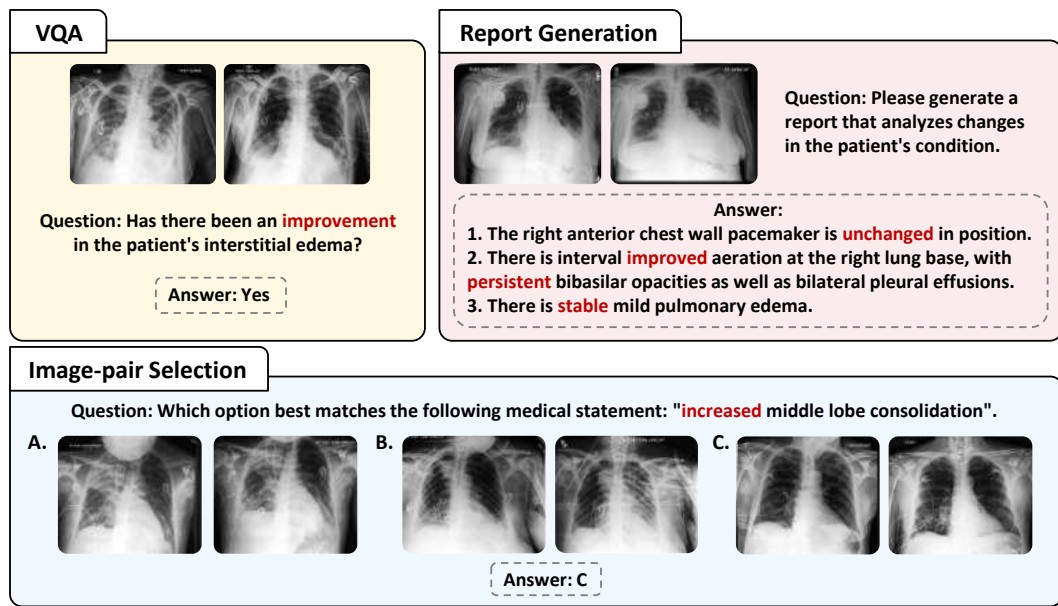

Figure 2: Examples of the three tasks in TEMMED-BENCH. Each question in these tasks is designed to challenge LVLMs' ability to analyze condition changes, providing a comprehensive evaluation of their temporal medical image reasoning ability.

describe the condition, we refer to it as a *single-visit description sentence*. In contrast, if a sentence contains phrases such as "*persistent* atelectasis", "*worsening* atelectasis", or "atelectasis has *improved*", where the adjectives indicate a change in condition, we refer to it as a *condition change description sentence*. Obviously, reports with condition change description sentences reflect that the doctor considered both current and previous conditions, not just information from a single visit.

The raw data for our benchmark are collected from the CheXpert Plus dataset (Chambon et al., 2024). We first collect reports in which every sentence is a condition change description sentence. For implementation, we select a set of keywords that are commonly used to describe condition changes, such as *increase*, *worsen*, and *stable*. We then use regular expressions to identify sentences containing at least one of these keywords (in any tense) as condition change description sentences. Next, we collect reports in which every sentence is a condition change description sentence as our target reports, and retrieve the corresponding frontal view image for each report as the current condition image. After identifying these target reports, we leverage the *patient_report_date_order* attribute in the CheXpert Plus dataset to track the historical visits of the same patient for each report. For each target report, we consider the most recent prior visit as the most relevant historical reference, and select the frontal view image from that visit as the historical condition image. In this way, a total of 18,144 instances were collected, each consisting of a pair of images and a report, with each sentence in the report describing condition changes observed between the images. More details on the data collection method can be found in Appendix A.1, and further discussion is provided in Appendix C.2.

## 2.3 TASK DATA COLLECTION

We randomly selected 1,000 instances as the test set and used the remaining instances as the knowledge corpus. Based on the test set instances, we constructed three tasks: VQA, report generation, and image-pair selection. Since the raw data has already been formatted for report generation, no additional processing is required for this task. Examples of the three tasks are shown in Figure 2.

**VQA Task Construction**  As illustrated in § 2.1, our VQA task adopts a binary setting, with "yes" or "no" as answers. To construct the VQA data, we leverage GPT-4o (OpenAI, 2024) to rephrase each report into question-answer pairs, following Li et al. (2023); Zhang et al. (2024); Xia et al. (2024). Specifically, each report is first segmented into individual sentences. Since each sentence in the reports is a condition change description sentence, it can be rephrased as a question that asks whether the patient's condition has changed. Then, all of these sentences are fed into GPT-4o to create VQA data with "yes" or "no" answers. To balance answer distribution, we prompt GPT-4o to

generate roughly equal numbers of "yes" and "no" questions. Additionally, each instance is manually reviewed to ensure that the questions target condition changes and the answers align with the truth in the report. More details of the construction process and quality control are provided in Appendix A.2.

**Image-pair Selection Task Construction** Our image-pair selection task adopts a three-option setting, with "A", "B", or "C" as answers. Unlike conventional multiple-choice question-answering tasks (Lau et al., 2018; Zhang et al., 2024; Hu et al., 2024), where each question consists of a target image and several textual options, our image-pair selection task is more vision-centric, with the options being image pairs, and the model is asked to choose the image pair that best matches a target medical statement. This task requires the model to analyze three image pairs at a time, which demands a high level of multi-image processing and reasoning abilities. For the data construction of this task, we first select a set of keywords that are often used to describe condition changes ($KW_C$), e.g., *persist*, *improve*, and *decrease*, as well as a set of pathology keywords ($KW_P$) that frequently occur in the reports (Chambon et al., 2024), e.g., *atelectasis*, *edema*, and *effusion*. We observe that most statements describing condition changes in the reports match the following regular expression:

$$[KW_C]_{\text{(in any tense)}} + 0 \sim 4 \text{ attributives} + [KW_P]$$

For example, "*improving* mild cardiogenic *edema*" and "*decrease* in left pleural *effusion*". Using this regular expression, we can assign multiple condition change statements to each image pair. Subsequently, for each condition change statement in each image pair, we construct an image-pair selection sample, where the specific condition change statement serves as the target medical statement in the question, and the corresponding image pair serves as the correct option. To generate incorrect options, we randomly sample image pairs with the same pathology but reflect a different condition change, thereby ensuring that only one option matches the target medical statement. For more details on the keywords and construction examples, please refer to Appendix A.3.

## 3 MULTI-MODAL RETRIEVAL AUGMENTATION

### 3.1 PROBLEM FORMULATION

In the medical domain, medical reports serve as a commonly used knowledge corpus for *text-only retrieval augmentation* (Tao et al., 2024; Xia et al., 2024; 2025; Sun et al., 2025). Formally, given a query tuple $\boldsymbol{Query} = (\boldsymbol{i_h}, \boldsymbol{i_c}, \boldsymbol{q})$ composed of a historical image, a current image, and a textual question, the retriever $\mathcal{R}$ retrieves a set of relevant textual medical reports $\boldsymbol{T} = [\boldsymbol{t_1}, \boldsymbol{t_2}, ..., \boldsymbol{t_N}]$ from a knowledge corpus $\mathcal{C}$. The LVLM $\mathcal{M}$ then takes $(\boldsymbol{Query}, \boldsymbol{T})$ as input for answer generation.

In our work, we explore a more challenging yet promising setting of *multi-modal retrieval augmentation*. As discussed in § 2, we use the collected image pairs and their condition change reports as our knowledge corpus. Formally, given a $\boldsymbol{Query}$, the retriever $\mathcal{R}$ returns a set of relevant medical images and their corresponding reports $(\boldsymbol{I_h}, \boldsymbol{I_c}, \boldsymbol{T}) = [(\boldsymbol{i_{h_1}}, \boldsymbol{i_{c_1}}, \boldsymbol{t_1}), (\boldsymbol{i_{h_2}}, \boldsymbol{i_{c_2}}, \boldsymbol{t_2}), ..., (\boldsymbol{i_{h_N}}, \boldsymbol{i_{c_N}}, \boldsymbol{t_N})]$ from $\mathcal{C}$. $\mathcal{M}$ then takes $(\boldsymbol{Query}, \boldsymbol{I_h}, \boldsymbol{I_c}, \boldsymbol{T})$ as input and generates the final answer.

### 3.2 PAIRWISE IMAGE RETRIEVAL

Existing cross-modal retrieval methods typically focus on calculating feature similarity between the target image and the reports in the knowledge corpus (Tao et al., 2024; Xia et al., 2024; 2025; Sun et al., 2025). Given an image $\boldsymbol{i}$ and a report $\boldsymbol{t}$, the similarity score is computed as follows:

$$Score = \text{Sim}(\text{Enc}_i(\boldsymbol{i}), \text{Enc}_t(\boldsymbol{t})), \tag{1}$$

where $\text{Enc}_i$ and $\text{Enc}_t$ denote the image and text encoder models, respectively, and $\text{Sim}$ denotes cosine similarity. However, this method does not fit with TEMMED-BENCH, since the reports in TEMMED-BENCH describe the condition changes between the historical image and the current image. Therefore, the report feature does not align with a single image, but rather with an image pair.

For retrieval augmentation in TEMMED-BENCH, we aim to retrieve instances whose condition changes are similar to those of the target images. To retrieve higher-quality data, we propose a pairwise image retrieval method. Specifically, given a target image pair $(\boldsymbol{i_h}, \boldsymbol{i_c})$ and an instance $(\boldsymbol{i_h^*}, \boldsymbol{i_c^*}, \boldsymbol{t^*})$ from the knowledge corpus, the pairwise image similarity score is computed as follows:

$$Score = \text{Sim}(\text{Enc}_i(\boldsymbol{i_h}), \text{Enc}_i(\boldsymbol{i_h^*})) + \text{Sim}(\text{Enc}_i(\boldsymbol{i_c}), \text{Enc}_i(\boldsymbol{i_c^*})), \tag{2}$$

| Model | Params | VQA | | Report Generation | | | | Image Selection |
|-------|--------|-----|-----|------|---------|--------|------|-----------------|
| | | Acc [50] | F1 [50] | BLEU | ROUGE-L | METEOR | Avg. | Acc [33.3] |
| *Open-Source LVLMs* | | | | | | | | |
| LLaVA-Med * | 7B | 51.65 | 35.23 | 9.85 | 6.51 | 7.10 | 7.82 | / |
| HuatuoGPT-Vision * | 7B | 53.00 | 41.65 | 7.00 | 6.54 | 18.30 | 10.61 | 33.29 |
| HealthGPT * | 14B | 46.30 | 43.49 | 11.61 | 9.26 | 18.70 | 13.19 | 33.64 |
| Qwen2.5-VL | 7B | 59.90 | 57.60 | 12.13 | 10.46 | 18.34 | 13.64 | 33.87 |
| Llama3.2-Vision | 11B | 45.65 | 45.48 | 8.57 | 7.94 | 15.77 | 10.76 | 33.06 |
| LLaVA-OneVision | 7B | 63.90 | 62.12 | 5.18 | 6.07 | 13.10 | 8.12 | 32.83 |
| *Proprietary LVLMs* | | | | | | | | |
| Gemini 2.5 Flash | / | 47.30 | 47.30 | 20.23 | 14.19 | 22.79 | 19.07 | 39.33 |
| Claude 3.5 Sonnet | / | 69.90 | 69.49 | 17.17 | 14.01 | 24.91 | 18.70 | 33.53 |
| GPT 4o | / | 51.65 | 47.16 | 12.74 | 12.32 | 23.46 | 16.17 | 32.60 |
| GPT 4.1 | / | 57.90 | 57.51 | 9.81 | 11.67 | 22.6 | 14.69 | 35.38 |
| GPT o4-mini | / | 79.15 | 78.94 | 20.54 | 15.75 | 25.71 | 20.67 | 35.03 |
| GPT o3 | / | 64.40 | 64.40 | 16.99 | 13.71 | 25.77 | 18.82 | 38.05 |

Table 3: Evaluation results on TEMMED-BENCH in the closed-book setting. Highest and second-highest scores for each task are highlighted in red and blue , respectively. Models marked with superscript * indicate medical LVLMs. Square-bracketed subscripts denote random-guess scores.

where the two terms ensure that the historical and current images of the retrieved instance are similar to their counterparts in the target image pair, respectively, while their joint evaluation ensures that the condition changes are similar. Then, image pairs with high similarity scores, along with their corresponding reports, are used as the retrieved instances.

In contrast to the VQA and report generation tasks, the implementation of retrieval augmentation for the image-pair selection task requires some special handling. For more details, please refer to Appendix B.2.

## 4 EXPERIMENTS

### 4.1 EXPERIMENTAL SETUP

We evaluate 12 popular LVLMs on TEMMED-BENCH, comprising 6 proprietary and 6 open-source models. Among the open-source models, 3 are medical LVLMs.

- **Proprietary models**: GPT o3 (OpenAI, 2025b), GPT o4-mini (OpenAI, 2025b), GPT 4.1 (OpenAI, 2025a), GPT 4o (OpenAI, 2024), Claude 3.5 Sonnet (Anthropic, 2024), and Gemini 2.5 Flash (Google, 2025).

- **Open-source models**: LLaVA-Med (7B) (Li et al., 2023), HuatuoGPT-Vision (7B) (Chen et al., 2024), HealthGPT (14B) (Lin et al., 2025), Qwen2.5-VL (7B) (Team, 2025), Llama3.2-Vision (11B) (Meta AI, 2024), and LLaVA-OneVision (7B) (Li et al., 2024).

**Evaluation Setup** For VQA, we use accuracy and F1 score as metrics. For report generation, following Jing et al. (2018) and Xia et al. (2025), we use BLEU (Papineni et al., 2002), ROUGE-L (Lin, 2004), and METEOR (Banerjee & Lavie, 2005). For image-pair selection, accuracy is used. Detailed evaluation prompts for closed-book and RAG scenarios can be found in Appendix B.1.

### 4.2 MAIN RESULTS

The evaluation results in the closed-book setting are shown in Table 3. Our TEMMED-BENCH clearly reveals the limitations of current LVLMs in temporal medical image reasoning. Most LVLMs perform at around the random guess level in the VQA and image-pair selection tasks, and achieve relatively low average scores in the report generation task. Moreover, the results highlight several noteworthy observations which we further discuss below. Additional analysis can be found in Appendix C.3.

| Model | VQA | | Report Generation | | | | Image Selection |
|---|---|---|---|---|---|---|---|
| | Acc | F1 | BLEU | ROUGE-L | METEOR | Avg. | Acc |
| *Open-Source LVLMs* | | | | | | | |
| LLaVA-Med * | 51.65 | 35.23 | 9.85 | 6.51 | 7.10 | 7.82 | / |
| + Text-only RAG | $54.45_{+2.80}$ | $41.71_{+6.48}$ | 17.51 | 13.35 | 17.76 | $16.21_{+8.39}$ | / |
| + Multi-Modal RAG | $56.00_{+4.35}$ | $44.70_{+9.47}$ | 12.70 | 10.74 | 17.16 | $13.53_{+5.71}$ | / |
| HuatuoGPT-Vision * | 53.00 | 41.65 | 7.00 | 6.54 | 18.30 | 10.61 | 33.29 |
| + Text-only RAG | $59.50_{+6.50}$ | $52.50_{+10.85}$ | 8.12 | 7.53 | 20.63 | $12.09_{+1.48}$ | - |
| + Multi-Modal RAG | $61.75_{+8.75}$ | $56.73_{+15.08}$ | 9.50 | 9.07 | 21.81 | $13.46_{+2.85}$ | $32.02_{-1.27}$ |
| HealthGPT * | 46.30 | 43.49 | 11.61 | 9.26 | 18.70 | 13.19 | 33.64 |
| + Text-only RAG | $59.05_{+12.75}$ | $57.13_{+13.64}$ | 13.46 | 10.50 | 20.68 | $14.88_{+1.69}$ | - |
| + Multi-Modal RAG | $69.90_{+23.60}$ | $68.71_{+25.22}$ | 14.96 | 11.46 | 20.42 | $15.61_{+2.42}$ | $33.53_{-0.11}$ |
| Qwen2.5-VL | 59.90 | 57.60 | 12.13 | 10.46 | 18.34 | 13.64 | 33.87 |
| + Text-only RAG | $63.15_{+3.25}$ | $60.26_{+2.66}$ | 12.80 | 12.33 | 22.12 | $15.75_{+2.11}$ | - |
| + Multi-Modal RAG | $65.35_{+5.45}$ | $63.03_{+5.43}$ | 13.26 | 12.43 | 21.69 | $15.79_{+2.15}$ | $35.15_{+1.28}$ |
| Llama3.2-Vision | 45.65 | 45.48 | 8.57 | 7.94 | 15.77 | 10.76 | 33.06 |
| + Text-only RAG | $63.05_{+17.40}$ | $58.76_{+13.28}$ | 12.46 | 10.86 | 20.13 | $14.48_{+3.72}$ | - |
| + Multi-Modal RAG | $64.10_{+18.45}$ | $60.52_{+15.04}$ | 14.26 | 12.37 | 21.45 | $16.03_{+5.27}$ | $35.15_{+2.09}$ |
| LLaVA-OneVision | 63.90 | 62.12 | 5.18 | 6.07 | 13.10 | 8.12 | 32.83 |
| + Text-only RAG | $78.25_{+14.35}$ | $78.21_{+16.09}$ | 9.30 | 9.12 | 19.81 | $12.74_{+4.63}$ | - |
| + Multi-Modal RAG | $78.65_{+14.75}$ | $78.35_{+16.23}$ | 11.36 | 10.15 | 20.27 | $13.93_{+5.81}$ | $33.64_{+0.81}$ |
| *Proprietary LVLMs* | | | | | | | |
| Gemini 2.5 Flash | 47.30 | 47.30 | 20.23 | 14.19 | 22.79 | 19.07 | 39.33 |
| + Text-only RAG | $49.95_{+2.65}$ | $49.20_{+1.90}$ | 22.88 | 17.27 | 22.98 | $21.04_{+1.97}$ | - |
| + Multi-Modal RAG | $51.40_{+4.10}$ | $50.54_{+3.24}$ | 23.49 | 21.48 | 23.17 | $22.71_{+3.64}$ | $40.26_{+0.93}$ |
| Claude 3.5 Sonnet | 69.90 | 69.49 | 17.17 | 14.01 | 24.91 | 18.70 | 33.53 |
| + Text-only RAG | $66.25_{-3.65}$ | $65.50_{-3.99}$ | 20.58 | 16.73 | 27.27 | $21.53_{+2.83}$ | - |
| + Multi-Modal RAG | $74.15_{+4.25}$ | $73.95_{+4.46}$ | 22.35 | 17.83 | 26.44 | $22.21_{+3.51}$ | $37.35_{+3.82}$ |
| GPT 4o | 51.65 | 47.16 | 12.74 | 12.32 | 23.46 | 16.17 | 32.60 |
| + Text-only RAG | $60.10_{+8.45}$ | $59.57_{+12.41}$ | 21.06 | 17.17 | 26.74 | $21.66_{+5.48}$ | - |
| + Multi-Modal RAG | $64.85_{+13.20}$ | $64.42_{+17.26}$ | 23.79 | 19.14 | 26.80 | $23.24_{+7.07}$ | $34.69_{+2.09}$ |
| GPT 4.1 | 57.90 | 57.51 | 9.81 | 11.67 | 22.6 | 14.69 | 35.38 |
| + Text-only RAG | $58.50_{+0.60}$ | $58.30_{+0.79}$ | 17.07 | 15.57 | 27.48 | $20.04_{+5.35}$ | - |
| + Multi-Modal RAG | $58.60_{+0.70}$ | $58.55_{+1.04}$ | 19.03 | 16.56 | 28.07 | $21.22_{+6.53}$ | $33.87_{-1.51}$ |
| GPT o4-mini | 79.15 | 78.94 | 20.54 | 15.75 | 25.71 | 20.67 | 35.03 |
| + Text-only RAG | $81.80_{+2.65}$ | $81.78_{+2.84}$ | 24.77 | 19.42 | 27.99 | $24.06_{+3.39}$ | - |
| + Multi-Modal RAG | $77.80_{-1.35}$ | $77.69_{-1.25}$ | 24.30 | 19.06 | 26.46 | $23.27_{+2.61}$ | $34.69_{-0.34}$ |
| GPT o3 | 64.40 | 64.40 | 16.99 | 13.71 | 25.77 | 18.82 | 38.05 |
| + Text-only RAG | $66.65_{+2.25}$ | $66.60_{+2.20}$ | 18.89 | 15.33 | 26.84 | $20.35_{+1.53}$ | - |
| + Multi-Modal RAG | $65.65_{+1.25}$ | $65.60_{+1.20}$ | 18.72 | 15.20 | 27.16 | $20.44_{+1.61}$ | $39.56_{+1.51}$ |

Table 4: Evaluation results on TEMMED-BENCH with text-only and multi-modal retrieval augmentation (using top-1 retrieval). The highest and second-highest scores for each model in each task are highlighted in red and blue, respectively. Relative performance changes compared to the closed-book setting are shown as subscripts, with red indicating gains and blue indicating drops.

**Importance of Reasoning Ability**   Among these results, the reasoning model GPT o4-mini achieves outstanding performance. It achieves the highest performance in both VQA and report generation tasks, with its VQA accuracy surpassing that of the second-highest model by 10%, and also shows relatively high performance in the image-pair selection task. These results suggest that, for TEMMED-BENCH, advanced image reasoning is a key capability for effectively addressing these tasks. Given that changes in medical image features across different visits are often subtle, our findings further underscore the importance of fine-grained visual reasoning in this context.

**Comparison between Proprietary and Open-Source LVLMs**   By comparing the performance of proprietary and open-source LVLMs on TEMMED-BENCH, we observed that proprietary LVLMs consistently achieve the best results across all tasks. Notably, in report generation, all proprietary LVLMs outperform the best open-source models, likely due to their superior language organization. For the VQA task, some open-source LVLMs achieve comparable performance to several proprietary models, with LLaVA-OneVision reaching 63.9% accuracy and Qwen2.5-VL achieving 59.9%. However, there is still a gap compared to GPT o4-mini and Claude 3.5 Sonnet.

**Degradation of Medical LVLMs**   To our surprise, among open-source models, medical LVLMs do not show better performance compared to general-domain LVLMs, with the report generation task and image-pair selection task achieving nearly the same performance on average, and even worse

| Retrieval Method | Acc | F1 |
|---|---|---|
| *Text-only RAG* | | |
| Image-to-Text | 58.90 | 56.77 |
| Image-to-Image | 58.15 | 55.95 |
| Pairwise Image | **59.05** | **57.13** |
| *Multi-Modal RAG* | | |
| Image-to-Text | 65.75 | 64.19 |
| Image-to-Image | 68.45 | 67.17 |
| Pairwise Image | **69.90** | **68.71** |

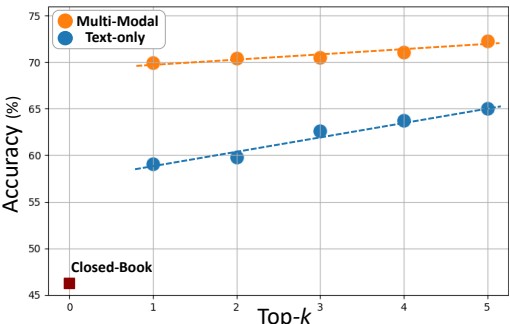

Table 5: Ablation study. We evaluate the performance of HealthGPT with different retrieval methods, including image-to-text, image-to-image, and pairwise image retrieval. The pairwise image retrieval method demonstrates the best performance among these methods.

Figure 3: Results of top-1 to top-5 retrieval augmentation HealthGPT. The orange line indicates multi-modal retrieval augmentation, while the blue line indicates text-only retrieval augmentation. The red square shows model performance without retrieval augmentation.

performance on the VQA task. A similar observation was also reported by Hu et al. (2024). Such findings reflect the lack of robustness and generalizability of current medical LVLMs, and reveals that prevailing medical knowledge fine-tuning schemes often erode the broad reasoning abilities inherited from general-domain pre-training. This underscores the need to develop adaptation frameworks that preserve general reasoning capabilities while reliably incorporating domain expertise.

## 4.3 MAIN RESULTS WITH RETRIEVAL-AUGMENTATION

The evaluation results in Table 4 demonstrate that multi-modal retrieval augmentation generally yields greater performance improvements across most models compared to text-only retrieval augmentation. Notably, compared to their text-only counterparts, HealthGPT, Claude 3.5 Sonnet, and GPT-4o demonstrate substantial gains in the multi-modal setting, with increases in VQA accuracy of 10.85%, 7.90%, and 4.75%, and improvements in report generation average score of 0.73, 0.68, and 1.59, respectively. Additional noteworthy observations are as follows.

**Comparison between Open-Source and Proprietary LVLMs with Retrieval Augmentation.** We noticed that open-source LVLMs exhibit a notably high performance gain in the VQA task after using retrieval augmentation. Specifically, HealthGPT, Llama3.2-Vision and LLaVA-OneVision show an increase of 23.60%, 18.45%, and 14.75% in VQA accuracy under multi-modal retrieval augmentation, respectively, compared to the closed-book setting. This makes some open-source LVLMs perform competitively with proprietary LVLMs in VQA. However, the performance of open-source LVLMs in the report generation and image-pair selection tasks still lags behind that of proprietary LVLMs, indicating that proprietary LVLMs still have an advantage in tasks requiring advanced language organization skills and strong multi-image processing capabilities.

**Challenges of Leveraging Retrieved Information in the Image-pair Selection Task** Although most LVLMs exhibit significant performance gains with retrieval augmentation, the improvement is less pronounced in the image-pair selection task, with several LVLMs even exhibiting a decrease in performance. We argue that this highlights the unique challenges of leveraging retrieved information in this specific task. Unlike the other two tasks, which involve only a single target image pair and require the model merely to compare the retrieved information with that pair to determine its usefulness, the image-pair selection task involves three target pairs. Consequently, the model must align the retrieved information with three separate pairs, splitting its attention and reconciling potentially conflicting cues, which substantially amplifies retrieval noise. The relatively complicated logic for retrieval augmentation in this setting underscores the need for advanced methods to improve LVLMs' ability to utilize retrieved information in such multifaceted scenarios.

## 4.4 ANALYSIS OF MULTI-MODAL RETRIEVAL AUGMENTATION

To further investigate the optimal settings for multi-modal retrieval augmentation, we use HealthGPT on the VQA task as an illustrative case. Additional experiments on proprietary model GPT 4o are presented in Appendix C.4.

**Ablation Study on Retrieval Methods**  Table 5 presents the ablation studies comparing the proposed pairwise image retrieval method with conventional image-to-text and image-to-image retrieval methods. Given a target image pair $(i_h, i_c)$ and an instance from the knowledge corpus $(i_h^*, i_c^*, t^*)$, the image-to-text retrieval computes the similarity score between $i_c$ and $t^*$, whereas the image-to-image retrieval computes the similarity score between $i_c$ and $i_c^*$. Results indicate that pairwise image retrieval achieved the highest performance, primarily due to two factors. First, for image-to-text retrieval, the report feature in TEMMED-BENCH does not correspond to a single image, as the report describes condition changes between two images. Consequently, directly calculating the feature similarity between the report and a single image introduces bias. Second, image-to-image retrieval relies solely on the similarity of current-visit images, ensuring similarity in current conditions between the target and retrieved instances, but does not guarantee similarity in condition changes. Therefore, in TEMMED-BENCH, only by considering both historical and current images during retrieval and ensuring similarity in both, can retrieved instances reflect similar condition changes.

**Impact of Top-$k$ Retrieval Augmentation**  To further analyze the impact of varying top-$k$ retrievals on augmentation performance, we conducted experiments to evaluate the performance of LVLMs under top-1 to top-5 retrieval augmentation settings, as shown in Figure 3. Notably, multi-modal retrieval augmentation consistently outperforms text-only retrieval augmentation across top-1 to top-5 settings, further confirming the effectiveness of incorporating multi-modal retrieved information in enhancing LVLM performance in the medical domain.

## 5 RELATED WORK

### 5.1 MEDICAL VISION-LANGUAGE BENCHMARKS

Medical Vision-Language Models (Med-LVLMs) have recently shown great promise in medical diagnostics, prompting interest in developing more advanced models (Li et al., 2023; Chen et al., 2024; Lin et al., 2025). For their evaluation, early benchmarks such as VQA-RAD (Lau et al., 2018), SLAKE (Liu et al., 2021), and PathVQA (He et al., 2020) focused on visual question-answering (VQA) but are limited by modality diversity and dataset size. More recent efforts (Zhang et al., 2024; Chen et al., 2024; Hu et al., 2024) have introduced larger VQA datasets with varied modalities. In addition, some studies (Luo et al., 2024; Bustos et al., 2020; Demner-Fushman et al., 2015; Chambon et al., 2024; Johnson et al., 2019) have built benchmarks for the report generation task, which present greater challenges for models in terms of long-form language generation, going beyond simple, short-sentence responses (Jing et al., 2018; Chen et al., 2020). However, these benchmarks all focus on analyzing a patient's condition based on a single-visit image. Some of the most recent works have begun to focus on LVLMs' ability to reason over multiple images (Yu et al., 2025; Mu et al., 2025; Yang et al., 2025), but they either rely on data sources that have not undergone reliable verification (Yu et al., 2025) or suffer from limitations in the comprehensiveness of their task design and evaluation (Mu et al., 2025; Yang et al., 2025). In contrast, TEMMED-BENCH provides a more comprehensive evaluation of LVLMs' ability to reason over temporal medical images.

### 5.2 MULTI-MODAL RETRIEVAL AUGMENTATION

Retrieval-Augmented Generation (RAG) has been proposed as an effective approach to address the inherent limitations of language models (Lewis et al., 2020). In the general domain, multi-modal knowledge retrieval has been widely studied to enhance generative models (Chen et al., 2022; Liu et al., 2023; Zhao et al., 2023; Yasunaga et al., 2023; Cui et al., 2024; Sharifymoghaddam et al., 2025). Recent benchmarks designed to evaluate multi-modal retrieval augmentation (Hu et al., 2025; Liu et al., 2025) further highlight the value of visual information retrieval for vision-centric tasks. However, in the medical domain, existing works mainly focus on performing retrieval augmentation using text-only information (Tao et al., 2024; Kumar & Marttinen, 2025; Sun et al., 2025; Xia

et al., 2024; 2025). Whether multi-modal retrieval augmentation is useful in this domain remains unexplored. Our work addresses this gap.

## 6 CONCLUSION

In this work, we introduce TEMMED-BENCH, a benchmark specifically designed to evaluate LVLMs' ability to reason over temporal medical images, by letting LVLMs track changes in patients' conditions between different clinical visits. We extend our benchmark to three tasks and release a knowledge corpus with over 17,000 instances. Through the evaluation of six proprietary and six open-source LVLMs, our findings highlight the limitations of current LVLMs in performing temporal reasoning with medical images. Furthermore, we investigate and validate the potential of multi-modal retrieval augmentation in the medical domain, emphasizing the efficacy of leveraging multi-modal information retrieval to enhance the performance of Med-LVLMs. Our work provides an evaluation that better reflects real-world clinical practice, guiding Med-LVLM development toward actual clinical needs.

### ETHICS STATEMENT

We used LLMs to assist in improving grammar, clarity, and wording in parts of this work. We also used LLMs to generate data only by perturbing or converting existing text, and all LLM outputs were human-checked. The models were not used to create new data. Therefore, these uses do not raise trustworthiness concerns. Apart from the above, all ideas, analyses, and conclusions were developed solely by the authors.

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

# A   DATA COLLECTION DETAILS

## A.1   RAW DATA COLLECTION

For raw data collection, we select a set of keywords commonly used to describe condition changes and then apply regular expressions to identify sentences that contain at least one of these keywords (in any tense) as condition change description sentences.

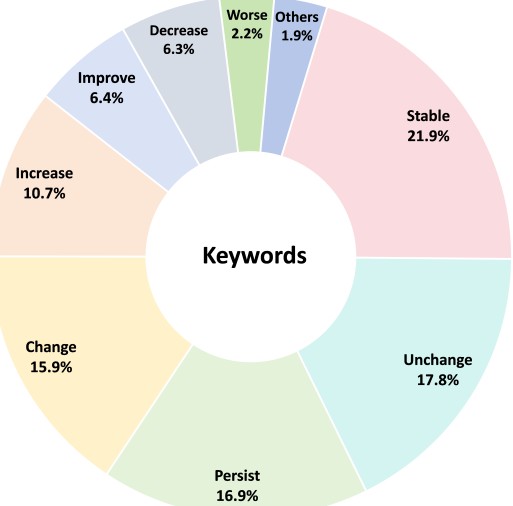

Figure 4: Keywords distribution.

The selected keywords are: [*"stable", "unchange", "change", "persist", "increase", "decrease", "improve", "worsen", "thicken", "thin", "progress", "deteriorate", "reduce", "resolve", "exacerbate"*]. The distribution of keywords in the collected instances is shown in Figure 4.

## A.2   VQA DATA CONSTRUCTION

**Construction Process**   To construct VQA data, we use GPT-4o to rephrase each sentence in our selected reports into a question-answer pair. The following is the prompt for this process, with {ANSWER} set to *Yes* for half of the data and *No* for the other half. Table 6 shows an example of how a report is converted into multiple VQA items.

---

**Prompts for VQA Data Construction**

You are a professional medical expert. I will provide you with a sentence from a medical report. Please generate a question with the answer '{ANSWER}' based on the provided sentence.

The question should focus on the content that indicates a change in the patient's condition. The subject of the question should be the medical image or the patient, not the report. Please include only the question and answer in your response.

Below are the given sentence: {SENTENCES}

---

---

**Report:**

1. Lines and tubes unchanged. 2. Stable pulmonary edema. 3. Small, slightly decreased aeration of the right lung. 4. Increased right pleural effusion is possible. 5. Increased thickening of the left parietal pleura, consistent with increased effusion, possibly loculated.

**Sentences:**

[sentence 1] Lines and tubes unchanged.

[sentence 2] Stable pulmonary edema.

[sentence 3] Small, slightly decreased aeration of the right lung.

[sentence 4] Increased right pleural effusion is possible.

[sentence 5] Increased thickening of the left parietal pleura, consistent with increased effusion, possibly loculated.

**Question-Answer Pair:**

**[QA 1]**

Q: Were there any changes in the placement of lines and tubes? | A: No.

**[QA 2]**

Q: Is the pulmonary edema in the patient stable? | A: Yes.

**[QA 3]**

Q: Has the aeration of the right lung increased? | A: No.

**[QA 4]**

Q: Has the right pleural effusion decreased? | A: No.

**[QA 5]**

Q: Have the medical images revealed an increased thickening of the left parietal pleura? | A: Yes.

---

Table 6: An example of VQA data construction

**Quality Control**   As our dataset for the VQA task is constructed using AI tools, we perform human evaluation to ensure its quality. For each question-answer pair, we manually assess whether the question targets a condition change and whether the answer is consistent with the ground truth described in the report. We find that nearly all answers are consistent with the report. However, 214 out of 2,000 questions (10.7%) do not target a condition change. We manually correct these question-answer pairs.

### A.3   IMAGE-PAIR SELECTION DATA CONSTRUCTION

For image-pair selection data construction, the keywords for describing condition changes ($KW_C$) and the pathology keywords ($KW_P$) are listed in Table 7. The selection of $KW_P$ is based on the frequently occurring pathology labels in the reports (Chambon et al., 2024).

Figure 5 shows an example of how to construct data for the image-pair selection task. We use regular expressions to associate medical statements with each image pair. The incorrect options are selected from image pairs with the same pathology but different changes in condition.

> **Regular Expression:** $[KW_C]_{\text{(in any tense)}} + 0 \sim 4$ attributives $+ [KW_P]$
>
> $KW_C$ : ["stable", "unchange", "persist", "increase", "decrease", "improve", "worsen", "thicken", "thin", "progress", "deteriorate", "reduce", "resolve", "exacerbate"]
>
> $KW_P$ : ["atelectasis", "cardiomegaly", "consolidation", "edema", "cardiomediastinum", "fracture", "lesion", "opacity", "effusion", "pneumonia", "pneumothorax"]

Table 7: Keyword selection for image-pair selection data construction

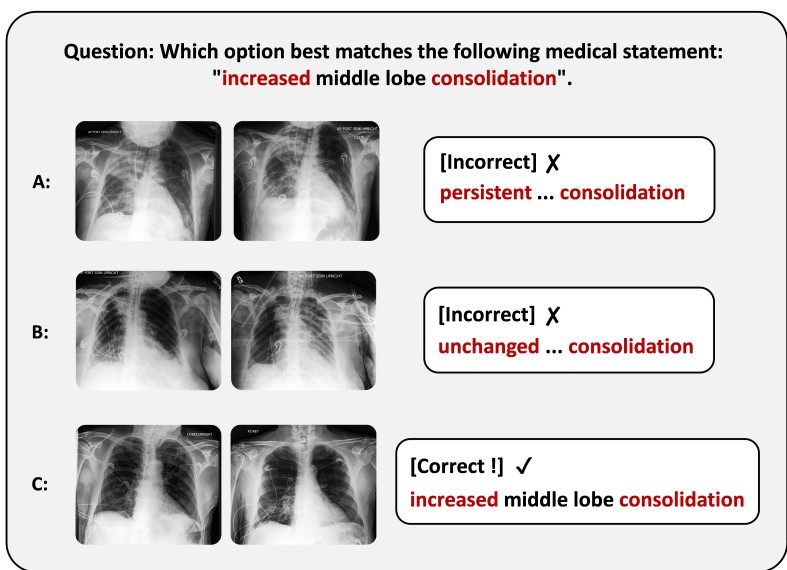

Figure 5: An example of image-pair selection data construction

## B  EXPERIMENT SETTING DETAILS

### B.1  EVALUATION PROMPTS

The following are the evaluation prompts for TEMMED-BENCH, where <image> denotes the image placeholder.

---

**Prompts for VQA Task (Closed-book)**

Last visit image: <image>

Current visit image: <image>

You are a professional radiologist. You are provided with two X-ray images from the same patient. The first image is from the last visit, and the second image is from the current visit.

I will ask you a question about the change in condition between the last and current visit of this patient. Please answer the question based on the two images and choose from the following two options: [Yes, No]. Please only include your final choice of 'Yes' or 'No' in your response.

Question: {VQA_QUESTION}

---

---

**Prompts for VQA Task (Text-only RAG)**

You are a professional radiologist. You are provided with a reference X-ray image report:

Report: {RAG_REPORT}

Please learn how to track changes in the patient's condition based on this example.

Now, you are given two new X-ray images from another patient:

Last visit image: <image>

Current visit image: <image>

The first new image is from the last visit, and the second new image is from the current visit. I will ask you a question about the change in condition between the last and current visit of this patient. Note that the diagnostic information from the reference report should not be directly used for diagnosis but only as a reference for comparison.

Please answer the question based on the two new images and choose from the following two options: [Yes, No]. Please only include your final choice of 'Yes' or 'No' in your response.

Question: {VQA_QUESTION}

---

**Prompts for VQA Task (Multi-Modal RAG)**

You are a professional radiologist. You are provided with two reference X-ray images from the same patient, along with the corresponding report for the current visit image:

Last visit image: <image>

Current visit image: <image>

Report for the current visit image: {RAG_REPORT}

Please learn how to analyze X-ray images and track changes in the patient's condition based on this example.

Now, you are given two new X-ray images from another patient:

Last visit image: <image>

Current visit image: <image>

The first new image is from the last visit, and the second new image is from the current visit. I will ask you a question about the change in condition between the last and current visit of this patient. Note that the diagnostic information from the reference images and report should not be directly used for diagnosis but only as a reference for comparison.

Please answer the question based on the two new images and choose from the following two options: [Yes, No]. Please only include your final choice of 'Yes' or 'No' in your response.

Question: {VQA_QUESTION}

---

**Prompts for Report Generation Task (Closed-book)**

Last visit image: <image>

Current visit image: <image>

You are a professional radiologist. You are provided with two X-ray images from the same patient. The first image is from the last visit, and the second image is from the current visit.

Please generate a report for the current visit image. You should consider the last visit image to analyze the changes in the patient's condition in your report. Please only include the content of the report in your response.

---

---

**Prompts for Report Generation Task (Text-only RAG)**

You are a professional radiologist. You are provided with a reference X-ray image report:

Report: {RAG_REPORT}

Please learn how to track changes in the patient's condition and generate reports based on this example.

Now, you are given two new X-ray images from another patient:

Last visit image: <image>

Current visit image: <image>

The first new image is from the last visit, and the second new image is from the current visit. Please generate a report for the new current visit image. You should consider the new last visit image to analyze the changes in the patient's condition in your report.

Note that the diagnostic information from the reference report should not be directly used for diagnosis but only as a reference for comparison. Please only include the content of the report in your response.

---

**Prompts for Report Generation Task (Multi-Modal RAG)**

You are a professional radiologist. You are provided with two reference X-ray images from the same patient, along with the corresponding report for the current visit image:

Last visit image: <image>

Current visit image: <image>

Report for the current visit image: {RAG_REPORT}

Please learn how to analyze X-ray images, track changes in the patient's condition, and generate reports based on this example.

Now, you are given two new X-ray images from another patient:

Last visit image: <image>

Current visit image: <image>

The first new image is from the last visit, and the second new image is from the current visit. Please generate a report for the new current visit image. You should consider the new last visit image to analyze the changes in the patient's condition in your report.

Note that the diagnostic information from the reference images and report should not be directly used for diagnosis but only as a reference for comparison. Please only include the content of the report in your response.

---

**Prompts for Image-pair Selection Task (Closed-book)**

A:

Last visit image: <image>

Current visit image: <image>

B:

Last visit image: <image>

Current visit image: <image>

C:

Last visit image: <image>

Current visit image: <image>

You are a professional radiologist. You are provided with three pairs of X-ray images. Each pair contains two X-ray images from the same patient. The first image in each pair is from the last visit, and the second one is from the current visit. Your task is to choose one of the options, based on the condition change from the last to the current visit, that best matches the following medical statement: '{MEDICAL_STATEMENT}'.

Please provide your answer by selecting the corresponding letter from the given choices. Please provide your final answer in the format: 'My answer is [option]' at the end of your response.

---

**Prompts for Image-pair Selection Task (Multi-Modal RAG)**

You are a professional radiologist. You are provided with two reference X-ray images from the same patient, along with the corresponding report:

Last visit image: <image>

Current visit image: <image>

Report: {RAG_REPORT}

Please learn how to analyze X-ray images and track changes in the patient's condition based on this example.

Now, you are provided with three pairs of X-ray images:

A:

Last visit image: <image>

Current visit image: <image>

B:

Last visit image: <image>

Current visit image: <image>

C:

Last visit image: <image>

Current visit image: <image>

Each pair contains two X-ray images from the same patient. The first image in each pair is from the last visit, and the second one is from the current visit. Your task is to choose one of the options, based on the condition change from the last to the current visit, that best matches the following medical statement: '{MEDICAL_STATEMENT}'.

Please provide your answer by selecting the corresponding letter from the given choices. Please provide your final answer in the format: 'My answer is [option]' at the end of your response.

---

### B.2 RETRIEVAL AUGMENTATION FOR THE IMAGE-PAIR SELECTION TASK

For the image-pair selection task, the input format differs significantly from the other two tasks: there are three target image-pairs rather than one, and it is unknown which pair is the correct answer. This setting prevents us from using image features to retrieve instances that match the medical statement in the question. Therefore, we adopt a text-to-text retrieval approach. Specifically, we represent the medical statement in the question and each report in the knowledge corpus using their corresponding TF-IDF embeddings. The relevance score between the medical statement $med\_s$ and each report $t$ is then computed as follows:

$$Score = \mathrm{Sim}(\text{TF-IDF}(med\_s), \text{TF-IDF}(t)), \tag{3}$$

where TF-IDF denotes TF-IDF embedding function, and $\mathrm{Sim}$ denotes cosine similarity. The retrieved report, along with its corresponding historical image and current image, is then used as a retrieved instance.

Additionally, We argue that, in the image-pair selection task, text-only retrieval augmentation is not meaningful. In this setting, there are three target image-pairs, and the retrieved report simply describes a condition similar to that in the medical statement of the question. Without the images corresponding to the retrieved report, this report merely restates the information already present in the medical statement. As a result, the model cannot effectively make use of the retrieved report. Only when the corresponding images are provided can the model compare each target image-pair with the retrieved image-pair, thereby making meaningful use of the retrieved report.

# C ADDITIONAL DISCUSSION

## C.1 DISCUSSION ON THE TOP-1 RETRIEVAL HACK

| Benchmarks | Model | Report Gerneration | | | |
|---|---|---|---|---|---|
| | | BLEU | ROUGE-L | METEOR | Avg. |
| **IU-Xray** | MMed-RAG | 31.38 | 25.59 | 32.43 | 29.80 |
| | Top-1 Retrieval Hack | 31.61 | 26.68 | 31.79 | 30.03 |
| **MIMIC-CXR** | MMed-RAG | 23.25 | 12.34 | 20.47 | 18.69 |
| | Top-1 Retrieval Hack | 26.16 | 19.99 | 25.28 | 23.81 |
| **TEMMED-BENCH (Ours)** | HealthGPT (fine-tuned) | 25.34 | 27.45 | 26.45 | 26.41 |
| | Top-1 Retrieval Hack | 24.24 | 22.11 | 24.15 | 23.50 |

Table 8: Evaluation results of top-1 retrieval hack. For each benchmark, the higher score is highlighted in red , and the lower score is highlighted in blue .

The evaluation results of the top-1 retrieval hack are shown in Table 8. We report the score for the top-1 retrieval hack, where the top-1 retrieved report is used directly as the answer. We also evaluate the performance of the model, which has already been fine-tuned on each benchmark, in the text-only retrieval augmentation setting. For the IU-Xray (Demner-Fushman et al., 2015) and MIMIC-CXR (Johnson et al., 2019) benchmarks, we use the RAG-based model MMed-RAG (Xia et al., 2025), while for TEMMED-BENCH, we fine-tune HealthGPT (Lin et al., 2025) on our knowledge corpus and use it for experiments.

Experimental results indicate that, on IU-Xray and MIMIC-CXR, simply taking the top-1 retrieved report as the answer even outperforms the fine-tuned models in the retrieval-augmented setting. However, TEMMED-BENCH is more robust to this hack. We argue that this is because previous benchmarks are conducted in a single-visit image analysis setting, which place more emphasis on pattern recognition and matching. Therefore, the demand for reasoning based on retrieved information to arrive at the answer is not high. In contrast, TEMMED-BENCH, due to its emphasis on reasoning, encourages models to leverage the retrieved information to perform reasoning over images, rather than simply copying or rephrasing the retrieved information, making it more robust to this hack.

## C.2 DISCUSSION ON THE DATA COLLECTION

| Right Input | | Random Historical Image | | Random Current Image | |
|---|---|---|---|---|---|
| Acc | F1 | Acc | F1 | Acc | F1 |
| 79.15 | 78.94 | $59.20_{-19.95}$ | $58.74_{-20.20}$ | $54.15_{-25.00}$ | $53.97_{-24.97}$ |

Table 9: Evaluation results of GPT o4-mini under the random historical image and random current image settings. Relative performance changes compared to the right input setting are shown as subscripts, with blue indicating drops.

To ensure the effectiveness of our data collection method, we conduct two additional experiments. Given the strong performance of GPT o4-mini on the VQA task, we select this setting as our baseline. Specifically, in the *Random Historical Image* experiment, we replace the historical image in each data sample with a random image; in the *Random Current Image* experiment, we replace the current image in each data sample with a random image.

As shown in Table 9, experimental results demonstrate that either replacing the historical image or replacing the current image leads to a significant decrease in performance. The model's performance drops to less than 60%, which is close to the level of random guessing. This indicates that both the collected historical and current images are essential for the model to achieve optimal performance, suggesting that our data collection method is effective.

| | Retrieval Method | HealthGPT | | GPT 4o | |
|---|---|---|---|---|---|
| | | Acc | F1 | Acc | F1 |
| **Text-only RAG** | Image-to-Text | 58.90 | 56.77 | 57.90 | 57.46 |
| | Image-to-Image | 58.15 | 55.95 | 60.05 | 59.54 |
| | Pairwise Image | **59.05** | **57.13** | **60.10** | **59.57** |
| **Multi-Modal RAG** | Image-to-Text | 65.75 | 64.19 | 61.60 | 61.09 |
| | Image-to-Image | 68.45 | 67.17 | 64.35 | 63.81 |
| | Pairwise Image | **69.90** | **68.71** | **64.85** | **64.42** |

Table 10: Ablation study on GPT 4o and HealthGPT. The pairwise image retrieval method demonstrates the best performance on both models.

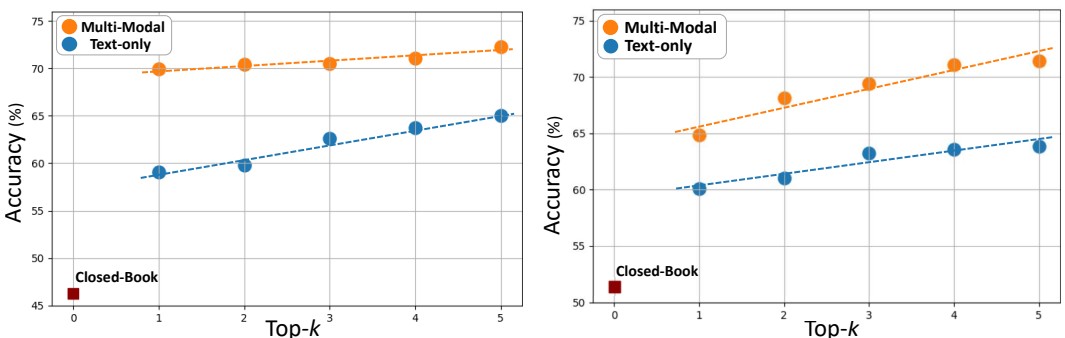

Figure 6: Results of top-1 to top-5 retrieval augmentation on HealthGPT (left) and GPT 4o (right). The orange line indicates multi-modal retrieval augmentation, while the blue line indicates text-only retrieval augmentation. The red square shows model performance without retrieval augmentation.

## C.3    DISCUSSION ON MODEL PERFORMANCE

**Low F1 Scores of LVLMs in the Closed-Book Setting**    For the evaluation results shown in Table 3, We noticed that the F1 score of LLaVA-Med is much lower than 50%, being only 35.23%. This is because LLaVA-Med tends to consistently output "Yes" when it is unable to answer a question, rather than randomly guessing between "Yes" and "No". As a result, the model's output exhibits a highly imbalanced class distribution, which leads to the low F1 score.

**Performance Gains of LVLMs with Retrieval Augmentation**    Based on the retrieval augmentation results in Table 4, we noticed that, for the VQA task, the performance gains of some open-source LVLMs are greater than those of proprietary LVLMs. Take HealthGPT and GPT-4o as examples. Although HealthGPT performs worse than GPT-4o under the closed-book setting, after adding Multi-Modal RAG, HealthGPT achieves 69.90% accuracy, even outperforming GPT-4o's 64.85% accuracy. We believe that this may be attributable to the limited ability of most LVLMs to analyze condition changes, whereas proprietary LVLMs tend to rely more on their own knowledge when incorporating retrieved information.

We evaluated the performance of the models when answering questions based solely on the retrieved information. More specifically, we prompted the models to answer the questions based only on the retrieved information without being given target images. We found that when they fully trust the retrieved information, HealthGPT and GPT-4o achieve 68.40% and 69.55% accuracy, respectively. These results indicate that GPT-4o refuses to trust the retrieved information for part of the questions in Multi-Modal RAG experiment, and since its own knowledge is limited in analyzing condition changes, this reliance on its own knowledge redirects GPT-4o to the wrong answer. In contrast, HealthGPT tends to rely more on the retrieved knowledge, leading to relatively higher performance. This observation further underscores the need to enhance LVLMs' capacity to analyze changes in the patients' conditions and to make informed judgments on whether to trust retrieved information or to rely on their own knowledge.

## C.4 ADDITIONAL ANALYSIS OF MULTI-MODAL RETRIEVAL AUGMENTATION

**Ablation Study on Retrieval Methods**   Table 10 presents ablation studies of different retrieval methods for two models. The results indicate that pairwise image retrieval achieves the highest performance on both the proprietary model GPT-4o and the open-source model HealthGPT.

**Impact of Top-$k$ Retrieval Augmentation**   Figure 6 shows the performance of GPT-4o and HealthGPT across top-1 to top-5 retrieval augmentation settings. For both models, multi-modal retrieval augmentation consistently outperforms text-only retrieval augmentation. Furthermore, by comparing the performance improvement from top-1 to top-5, we observe that GPT-4o demonstrates a significantly higher increase in accuracy (6.6%) compared to HealthGPT (2.37%). These results suggest that LVLMs with superior multi-image processing capabilities derive greater benefits from an increased number of retrieved instances, underscoring the importance of enhancing multi-image processing ability to fully leverage multi-modal retrieved information.

## C.5 LONG-TERM DIFFICULTY OF TEMMED-BENCH

| Model | Params | VQA | | Report Generation | | | | Image Selection |
|---|---|---|---|---|---|---|---|---|
| | | Acc [50] | F1 [50] | BLEU | ROUGE-L | METEOR | Avg. | Acc [33.3] |
| *Proprietary LVLMs* | | | | | | | | |
| Gemini 2.5 Pro | / | 50.35 | 49.70 | 22.42 | 17.31 | 27.83 | 22.52 | 40.02 |
| Claude Sonnet 4.5 | / | 66.10 | 65.90 | 9.93 | 11.22 | 21.64 | 14.26 | 33.64 |
| GPT 5.1 | / | 59.35 | 59.35 | 16.03 | 14.41 | 28.51 | 19.65 | 41.58 |

Table 11: Additional evaluation on three of the latest SOTA proprietary LVLMs.

To further assess the long-term difficulty of TEMMED-BENCH, we conducted an additional evaluation on three of the latest SOTA proprietary LVLMs, Gemini 2.5 Pro, Claude Sonnet 4.5, and GPT 5.1, as shown in Table 11. These results indicate that even the most advanced LVLMs still struggle on TEMMED-BENCH, confirming that it remains substantially challenging and leaves ample headroom for future LVLMs.

## C.6 DATA CONSTRUCTION AND HUMAN VALIDATION IN TEMMED-BENCH

To clearly discuss the faithfulness and validity of the annotations in our benchmark, we clarify below how our data construction pipeline is deliberately designed to minimize the need for additional expert re-labeling, while still maintaining high clinical fidelity.

**Report generation task**   For the report generation task, we directly select radiology reports from the CheXpert Plus dataset in which every sentence is a condition change description sentence, without generating new clinical findings. These reports are authored and validated in routine clinical workflow by radiologists, and our pipeline only selects those in which every sentence describes a change in the patient's condition across visits, without modifying their content. Thus, the correctness of the target reports is inherited from the original dataset, and no extra expert re-annotation is introduced by our benchmark.

**VQA task**   For the VQA task, we construct QA pairs by rephrasing each condition-change sentence in the target reports into a binary question whose answer is "yes" or "no". Concretely, we prompt an LLM to perform a purely syntactic transformation from the original sentence to a question, keeping the underlying clinical statement unchanged. For example, the sentence "Stable pulmonary edema" is converted into "Is the pulmonary edema in the patient stable?" with answer "Yes". This operation does not require any medical knowledge beyond preserving the meaning of the original sentence. Thanks to this data construction pipeline, human review of the data is quite simple and requires only very basic medical knowledge, such as familiarity with a limited number of diseases in the dataset. The human reviewer only needs to ensure that each question indeed targets the described change in

condition and that the answer is consistent with the corresponding sentence in the original report. This human review process for the 2,000 VQA examples was carried out over a two-week period by a single person who had prior knowledge of the diseases covered in the dataset.

**Image-pair selection task**    For the image-pair selection task, we deliberately avoid using LLMs to interpret reports. Instead, we design regular-expression patterns over a fixed set of change-keywords and pathology terms, tailored to the highly standardized wording of radiology reports. During development, we manually reviewed around 100 reports extracted by our patterns and confirmed that all of them contained the intended target condition change (precision = 100%). This pattern-based construction gives us a deterministic mapping from the report text to the label, avoiding the additional uncertainty that LLM-based interpretation would introduce and thereby reducing the need for radiologist re-verification.

## C.7  ROBUSTNESS OF THE REGULAR EXPRESSION-BASED EXTRACTION PROCEDURE

We provide below a clear discussion of the design of our regular-expression-based extraction procedure and its robustness.

**Variability of clinical language in chest X-ray reports**    Our regular expressions are designed based on a broad empirical examination of the clinical language used in chest X-ray reports in the CheXpert dataset. In practice, we found that these reports are written in a highly standardized, template-like style, with a small and constrained set of adjectives used to describe temporal evolution (e.g., stable, unchanged, improved, worsened, resolved). Prior work has also documented that radiology reporting is often highly standardized and relies on a controlled lexicon rather than open-ended narrative descriptions (Bosmans et al., 2012; Johnson et al., 2019; Yates et al., 2021). This constrained reporting style makes a compact set of regular expressions an effective and reliable extraction approach in this specific setting.

**High-precision focus of the regular expression design**    We adopt a regular expression-based approach primarily to guarantee that the extracted cases indeed contain the target condition changes, rather than to exhaustively capture all possible cases of that condition. In other words, our design prioritizes high precision over high recall. During development, we manually reviewed around 100 reports extracted by our patterns and confirmed that all of them contained the intended target condition change (precision = 100%).

## C.8  DATA SPLITTING AND THE POTENTIAL INFLATION OF RAG GAINS

In TEMMED-BENCH, the data split is performed at the instance level. As a result, cases from the same patient may appear in both the test set and the knowledge corpus. We would like to note that, in real-world clinical practice, it is reasonable to include records from other visits of the same patient in the retrieval pool, and it is often desirable to retrieve a patient's other visits to help interpret the current condition. Our benchmark design intentionally follows this realistic medical RAG scenario.

One concern about this split method is that, if patient overlap exists between the test set and the knowledge corpus, there might be near-duplicate visits from the same patient and the RAG gains might be inflated. To directly address this concern about possible inflation, we conducted additional analyses and ablations.

**1. How often does same-patient retrieval occur?**

In the VQA test set (2,000 cases), we find that only 185 cases ($\tilde{9}.2\%$) have a top-1 retrieved instance from the same patient as the target case.

**2. What happens if we forbid same-patient retrieval?**

We re-evaluated two representative models, HealthGPT and GPT-4o, by excluding all same-patient instances from the retrieval pool. The results are shown in Table 12.

The absolute changes for HealthGPT are around 0.3% in accuracy and 0.7% in F1. GPT-4o shows the same trend, with even smaller differences when instances from the same patient are excluded. These results indicate that allowing same-patient data in the retrieval pool does not materially inflate the

| Retrieval Setting | HealthGPT | | GPT 4o | |
|---|---|---|---|---|
| | Text-only RAG | Multi-Modal RAG | Text-only RAG | Multi-Modal RAG |
| | Acc / F1 | Acc / F1 | Acc / F1 | Acc / F1 |
| Same-patient allowed | 59.05 / 57.13 | 69.90 / 68.71 | 60.10 / 59.57 | 64.85 / 64.42 |
| Same-patient excluded | $58.85_{-0.2}$ / $56.83_{-0.3}$ | $69.25_{-0.65}$ / $67.99_{-0.72}$ | $60.05_{-0.05}$ / $59.52_{-0.05}$ | $64.60_{-0.25}$ / $64.21_{-0.21}$ |

Table 12: Experiments on RAG performance after forbidding same-patient retrieval.

RAG gains reported in our benchmark. A key reason is that TEMMED-BENCH focuses on condition changes rather than single-visit pattern matching: even if the retrieved image pair comes from the same patient and the absolute conditions of these images are likely to be similar, the condition changes are not guaranteed to be similar. Therefore, based on these results and this analysis, the concern that retrieving instances from the same patients would inflate the RAG gains does not appear to be supported in our benchmark.

## C.9 CASE ANALYSIS

---

**Case Analysis (VQA)**

**Last visit image**          **Current visit image**

- **Question:** Does the medical image show any change in the bibasilar opacities?

- **Ground Truth Answer:** No

- **LVLMs Answer:**

(✓) No: [GPT o4-mini] [GPT o3]

(✗) Yes: [GPT 4.1] [GPT 4.1] [GPT 4o] [Claude 3.5 Sonnet] [Gemini 2.5 Flash] [LLaVA-OneVision] [Llama3.2-Vision] [Qwen2.5-VL] [HealthGPT] [HuatuoGPT-Vision] [LLaVA-Med]

- **Analysis:**

For this VQA case, most models tend to respond "Yes", and only two advanced proprietary models answer it correctly. One possible reason is that the two CXRs differ in exposure, contrast, rotation, and inspiration level. These factors slightly alter the brightness at the lung bases, creating the illusion that the bibasilar opacities have changed, even though they are clinically stable. In addition, there are clear interval changes in other structures (e.g., lines and hardware), so the model "sees" change and answers "Yes" instead of restricting its judgment strictly to the bibasilar opacities.

---

## Case Analysis (Report Generation)

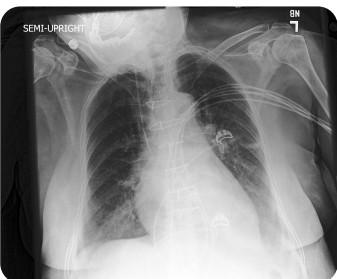 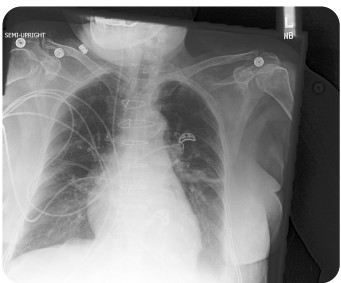

**Last visit image**   **Current visit image**

- **Ground Truth Report:**

1. Single frontal view of the chest demonstrates unchanged positioning of right ij central venous catheter.

2. Cardiomediastinal silhouette unchanged in size and configuration, with atherosclerotic calcifications at the arch.

3. Unchanged small/moderate sized left greater than right pleural effusions. Unchanged mild pulmonary edema. Unchanged slight linear opacity in the left midlung zone and retrocardiac left base, likely representing atelectasis.

- **[1] [LLaVA-Med]:**

The report for the current visit image should include an analysis of the changes in the patient's condition since the last visit. This may involve comparing the current image to the last visit image, identifying any new or worsening findings, and providing recommendations for further evaluation or treatment based on the observed changes.

- **[2] [HuatuoGPT-Vision]:**

The current visit chest X-ray image demonstrates several notable findings when compared to the last visit. The most significant change is an increase in opacification within both lung fields, suggesting progression of pulmonary infiltrates or edema.

1. Bilateral Pulmonary Opacities: There are diffuse bilateral hazy and patchy areas seen throughout both lungs on this semi-upright view. These opacities appear more extensive than those observed in the previous radiograph.

2. Possible Progression of Infiltrates/Edema: Compared with the prior study, there seems to be a worsening pattern of these abnormalities which could indicate ongoing inflammatory processes such as pneumonia, acute respiratory distress syndrome (ARDS), or other causes leading to increased fluid accumulation in alveolar spaces.

3. Cardiac Silhouette: While not clearly delineated due to overlying opacities, it appears that cardiac size remains stable without obvious signs of cardiomegaly at first glance; however, further evaluation may be necessary given the presence of widespread pulmonary changes.

4. Mediastinal Structures: No overt mediastinal shift or abnormal widening noted here either now or previously based upon available views.

5. Bones & Soft Tissues: Visible bony structures including ribs and clavicles show no new fractures or lytic lesions since the last examination. Soft tissues also do not reveal any remarkable pathology between visits.

In summary, while maintaining vigilance regarding potential underlying conditions causing progressive pulmonary infiltration or edema, clinical correlation with patient symptoms like dyspnea, cough production, fever history, etc., would aid significantly in refining differential diagnoses and guiding appropriate management strategies. Further investigations might include laboratory tests, CT scans, bronchoscopy depending on suspicion raised by these imaging findings. Close monitoring and follow-up studies will likely be required considering the apparent deterioration from the last visit's condition.

- **[3] [HealthGPT]:**

The current chest X-ray shows significant improvement compared to the previous image. The bilateral infiltrates have resolved, indicating a positive response to treatment. The lung fields are clear, and there are no signs of consolidation or pleural effusion. The heart size remains within normal limits, and the mediastinal structures are unremarkable. The bony structures, including the ribs and vertebrae, show no acute abnormalities. Overall, the patient's condition appears to have improved significantly.

• [4] [Qwen2.5-VL]:

The current chest x-ray shows no significant changes compared to the last visit. The heart size, lung fields, and mediastinal structures appear stable. There is no evidence of new or worsening pleural effusion, pneumothorax, or consolidation. The metallic foreign body in the right thoracic region remains unchanged in position. The patient's overall lung parenchyma appears clear without any new infiltrates or opacities. No acute changes are noted in the bony structures. The patient's condition seems stable based on this imaging.

• [5] [Llama3.2-Vision]:

The current X-ray image reveals a significant change in the patient's condition compared to the last visit. The lungs appear more opaque, indicating an increase in lung density, which may be due to fluid accumulation or inflammation. The heart size has increased, suggesting potential cardiac enlargement or fluid overload. The diaphragm appears elevated, which could be related to the increased lung density or other underlying conditions. The pleural spaces are clear, but there is a notable increase in the size of the pleural effusion on the left side, indicating fluid accumulation in the pleural space. The mediastinum appears normal, and there are no signs of pneumothorax or pneumomediastinum. The bones appear normal, with no fractures or other abnormalities noted. Overall, the current X-ray image suggests that the patient's condition has worsened, with possible fluid accumulation in the lungs and pleural space, and potential cardiac enlargement. Further evaluation and diagnostic tests are recommended to determine the underlying cause of these changes.

• [6] [LLaVA-OneVision]:

Based on the provided images, I observe several notable differences between the last visit and the current visit X-ray:

1. **Sternotomy Wound Closure**: In the current visit image, there is a visible surgical closure of the sternotomy wound, which appears to be intact and without signs of dehiscence or infection.

2. **Central Line**: The central line present in the last visit image is no longer visible in the current visit image, suggesting it has been removed.

3. **Pacemaker Lead**: The pacemaker lead that was partially visible in the last visit image is now fully visible and appears to be in place without any signs of migration or dislodgement.

4. **Ribs and Spine Alignment**: The ribs and spine appear to be aligned correctly in both images, with no obvious fractures or misalignments.

5. **Heart Size and Shape**: The heart size and shape seem consistent between the two images, without any apparent enlargement or abnormality.

6. **Lung Fields**: The lung fields are clear in both images, with no signs of consolidation, pneumothorax, or other abnormalities.

7. **Diaphragm**: The diaphragm is intact and shows no free air or fluid levels.

8. **Soft Tissue**: There are no visible soft tissue masses or abnormalities in the chest wall or neck region.

In summary, the current visit X-ray shows a stable condition with no new acute findings compared to the last visit. The surgical closure of the sternotomy wound, removal of the central line, and the position of the pacemaker lead are all within expected parameters for post-operative care.

• [7] [Gemini 2.5 Flash]:

Compared to the prior radiograph, there is interval worsening of bilateral airspace opacities, now with a diffuse pattern. This is consistent with worsening pulmonary edema or ARDS. The endotracheal tube tip remains approximately 3 cm above the carina. The nasogastric tube

tip is in the stomach. The right internal jugular central venous catheter tip is in the superior vena cava. The sternal wires and epicardial pacing wires remain in stable position. There is no pneumothorax or large pleural effusion.

• [8] [Claude 3.5 Sonnet]:

1. Stable appearance of a left-sided Port-a-Cath with catheter tip appropriately positioned in the superior vena cava.

2. No significant interval changes in the bilateral pulmonary parenchyma compared to prior study.

3. No evidence of pneumothorax or pleural effusion.

4. Heart size and mediastinal contours remain stable.

5. Multiple monitoring leads and EKG electrodes noted on the chest wall.

• [9] [GPT 4o]:

1. There is interval placement of additional medical devices since the prior study, with no evidence of malpositioning noted. All devices, including previously placed lines, remain appropriately positioned.

2. Persistent cardiomegaly with no interval resolution observed.

3. The previously noted pulmonary opacities remain visible in the current study, with no significant interval changes to suggest worsening or resolution. No new opacities are identified.

4. Mild interval progression of pleural effusion bilaterally, more pronounced on the right side.

5. No evidence of new pneumothorax or significant interval change in the aeration of observed lung fields.

Comparison with the prior study suggests a stable to mildly worsened condition, particularly with slight progression of pleural effusion. Continued monitoring is advised.

• [10] [GPT 4.1]:

1. **Stable cardiomediastinal silhouette** compared to the previous study. No significant interval change in cardiac or mediastinal contours.

2. **Persistent bilateral interstitial and alveolar opacities** are noted, with no significant interval worsening or improvement. These findings are compatible with ongoing pulmonary edema or infiltrative process.

3. **Stable position of previously seen sternal wires and prosthetic devices**. No new hardware or acute postsurgical complications identified.

4. **No evidence of new pneumothorax or large pleural effusion** compared to the prior exam.

5. **Lines and tubes** remain appropriately positioned.

6. **No acute bony abnormalities**.

Overall, there is no significant interval change in the radiographic findings compared to the previous study. Continued clinical correlation is recommended.

• [11] [GPT o4-mini]:

1. Interval slight progression of bilateral airspace opacities, most conspicuous in the left lower lung field, compatible with worsening pulmonary edema/ARDS or evolving pneumonia.

2. Endotracheal tube tip lies low, at approximately the T5–T6 level (near the carina); recommend withdrawal by 1–2 cm to optimize position.

3. Central venous and orogastric/NG tubes remain appropriately positioned and unchanged.

4. No new pneumothorax. Small bilateral pleural effusions persist without significant change.

5. Cardiomediastinal silhouette and osseous structures are stable.

• [12] [GPT o3]:

1. Since the prior study there is interval increase in basilar and perihilar parenchymal opacities with new-to-moderate bilateral pleural effusions (left > right). Findings most likely reflect progression of pulmonary edema and/or superimposed pneumonia; correlate clinically.

2. Heart size, mediastinal contours and postoperative sternotomy wires are unchanged.

3. Indwelling hardware and vascular lines remain in stable and satisfactory positions.

4. No evidence of pneumothorax or acute bony abnormality.

• **Analysis:**

For this report generation case, most models failed to accurately identify the conditions that should be included in the report. For open-source LVLMs, although they can generate long answers, most of the content does not target the key conditions that real clinicians want to analyze and is therefore useless. For proprietary LVLMs, some models successfully identified the desired conditions (as shown in blue), but many findings in the ground-truth report were still not covered. This indicates that enabling LVLMs to accurately identify the clinically relevant conditions that warrant analysis in real-world clinical scenarios remains a major challenge and deserves further investigation.

---

**Case Analysis (Image-pair Selection)**

A
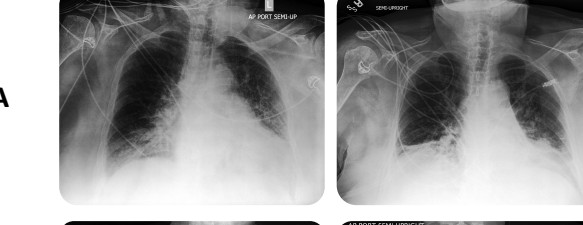

B
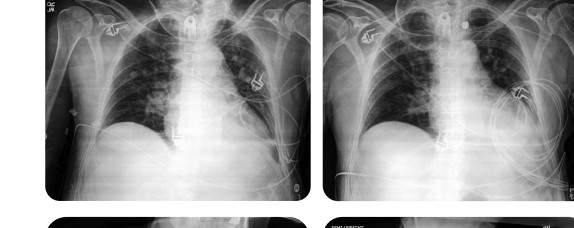

C
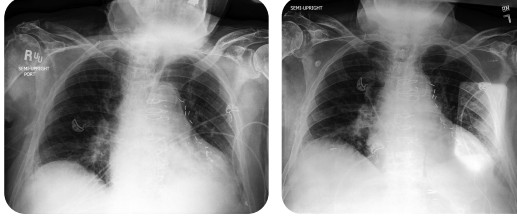

**Last visit image**     **Current visit image**

• **Question:** Your task is to choose one of the options, based on the condition change from the last to the current visit, that best matches the following medical statement: 'unchanged basilar atelectasis'.

• **Ground Truth Answer:** C

• **LVLMs Answer:**

(✗) A: [GPT o4-mini] [Gemini 2.5 Flash] [Qwen2.5-VL]

(✗) B: [GPT 4o] [LLaVA-OneVision] [HealthGPT] [HuatuoGPT-Vision]

(✓) C: [GPT o3] [GPT 4.1] [Claude 3.5 Sonnet] [Llama3.2-Vision]

• **Analysis:**

For this image-pair selection case, only four LVLMs answered correctly. Considering the low accuracy among all the LVLMs we evaluated on this task, we believe that the main bottleneck

in their performance is still their limited ability to process multiple images. We highlight enhancing LVLMs' multi-image processing ability in the medical domain as an important direction for boosting model performance in real-world clinical scenarios.

## C.10 LIMITATIONS AND FUTURE DIRECTIONS

**Limited modality diversity** TEMMED-BENCH is currently constructed solely from chest X-ray data, which constrains its direct applicability to other modalities such as CT, MRI, or ultrasound. Our design goal is to isolate temporal medical image reasoning in a modality-agnostic way, and chest X-ray offers large-scale longitudinal studies with reports that explicitly describe interval changes, making it a practical starting point. Extending our benchmark to additional imaging modalities will require access to suitably curated longitudinal datasets and is an important direction for future work.

**Two-timepoint temporal setting** TEMMED-BENCH focuses on temporal reasoning between two visits, rather than full longitudinal trajectories with three or more time points. While this two-timepoint formulation already captures a core component of temporal reasoning, it does not fully reflect more complex clinical scenarios involving longer follow-up sequences. Developing sequence-level benchmarks with richer longitudinal trajectories is a key avenue worth exploring in the future.

