# OpenReview forum: "TemMed-Bench: Evaluating Temporal Medical Image Reasoning in Vision-Language Models"
_ICLR.cc/2026/Conference — Submitted to ICLR 2026_

### Official Review · Reviewer_sLS7 · 2025-10-24

**Soundness:** 3
**Presentation:** 3
**Contribution:** 3
**Rating:** 6
**Confidence:** 5

**Summary:**

The paper introduces TEMMED-BENCH, a benchmark for assessing temporal medical image reasoning for LVLMs. Each test item pairs a historical and a current chest radiograph from the same patient and evaluates models on three tasks: binary VQA about condition change, change-focused report generation, and an image-pair selection task where the model must pick the pair that matches a specified change statement. The paper also proposes pairwise image retrieval for multi-modal RAG, which retrieves image-pairs (and their reports) whose historical/current images are jointly similar to the query pair. The result shows that several LVLMs benefit more from multi-modal RAG than text-only RAG. Overall, most LVLMs perform near chance on several tasks; reasoning-oriented proprietary models fare best, but none are strong yet on temporal change analysis.

**Strengths:**

- The motivation is crisp and clinically grounded: real radiology practice is often longitudinal, while most benchmarks are single-visit.
- The proposed image-pair selection task is novel to me. It is also more vision-centric than typical multi-choice VQA and stresses multi-image reasoning.
- The paper is clear, with concrete task definitions, corpus statistics, and straightforward evaluation protocols.

**Weaknesses:**

**1. Some claims are to be clarified with discussion**

- Please check the claim “the first benchmark that focuses on evaluating the temporal reasoning ability of LVLMs”, since there are existing works (e.g., MedFrameQA [1], MMXU [2], MedMIM [3], ICG-CXR [4]) that also feature in temporal evaluation by gathering longitudinal studies from a pool of imaging studies). The authors should discuss how their work differ from those.

[1] MedFrameQA: A Multi-Image Medical VQA Benchmark for Clinical Reasoning (arxiv 2025.05)

[2] MMXU: A Multi-Modal and Multi-X-ray Understanding Dataset for Disease Progression (ACL 2025)

[3] Medical Large Vision Language Models with Multi-Image Visual Ability (MICCAI 2025)

[4] Towards interpretable counterfactual generation via multimodal autoregression (MICCAI 2025)


- In Line 201, the authors claim that “We randomly selected 1,000 instances as the test set and used the remaining instances as the knowledge corpus.” I am unclear whether the split is based on patients or instances (the images from a same patient can appear in multiple instances). If patient overlap exists between test set and knowledge corpus, there might be near-duplicate visits from the same patient and the RAG gains might be inflated. I suggest the authors check the split and see if the results need to be re-evaluated.


**2. Some important details are missing**
- During dataset curation, from multiple longitudinal images from the same patient, how are each image pairs in each case chosen? Are these image pairs from consecutive examinations (e.g., study1-study2, study3-study4), from non-consecutive examinations (e.g., study1-study3, study2-study4), or from both?
- What is the distribution of time gaps between historical/current images, and how do performance and error types vary by short vs. long intervals?

- While the image-pair selection task is novel to me, I am curious how this task is done by existing LVLMs since the models may not natively support this task. Specifically, how do the authors modify the LVLMs so they can percept the order of input image-pairs (“A”, “B”, “C”)? Do the authors directly concatenate tokens of different image pairs together? Does that need an extra level of order position encoding?

- In Lines 223-226, could the authors provide some statistics that leads to this observation? (e.g. how many reports match the regular expression) This can benefit follow-up research on CXR report analysis.

- Which image encoders are used in the pairwise retrieval score?

**Questions:**

Please see "Weaknesses"

---

> ### Author Response · Authors · 2025-11-20
>
> Thank you for your detailed and thoughtful feedback. We greatly appreciate your positive assessment of our work's quality and clarity. Below, we address the weaknesses and questions you raised and clarify points that may have been unclear.
>
> ---
>
>
> **[Weakness 1.1] Comparison with Existing Work**
>
>
> We thank the reviewer for pointing out these recent datasets that also involve temporal evaluation. We acknowledge that these works already include temporal reasoning components, and we apologize for having missed them in our literature review. **Our original wording, “The first benchmark that focuses on evaluating the temporal reasoning ability of LVLMs,” is indeed too strong, and we will soften it to “The first multi-task benchmark that focuses on comprehensively evaluating the ability of LVLMs to perform temporal reasoning on medical images in both closed-book and open-book settings.”** and clarify the differences from these concurrent works in the revised version. Below, we summarize the key differences between TemMed-Bench and these datasets.
>
>
> **MedFrameQA [1]** proposes a multi-image medical VQA benchmark constructed from temporally coherent frames in medical education videos collected from YouTube. In contrast, TemMed-Bench is built from de-identified longitudinal studies in a hospital setting, which have undergone more reliable verification and more closely reflect real clinical follow-up workflows. Moreover, MedFrameQA adopts a single, closed-ended VQA task, while our benchmark contains three complementary tasks, including VQA, change-focused report generation, and image-pair selection, providing a more comprehensive evaluation of temporal reasoning. Finally, our work further studies temporal reasoning under retrieval-augmented settings, which is not covered in MedFrameQA.
>
> **MMXU [2]** is a MedVQA dataset that focuses on identifying disease progression between two chest X-rays and introduces MedRecord-Augmented Generation based on textual historical records. While MMXU is highly relevant in emphasizing temporal comparison, it still follows a VQA-only evaluation protocol with a limited answer space, whereas TemMed-Bench additionally evaluates open-ended change-report generation and an image-centric selection task. Furthermore, MMXU explores text-only augmentation, while our study investigates multimodal (image + text) retrieval-augmented temporal reasoning. In addition, we conduct a more comprehensive evaluation of state-of-the-art proprietary LVLMs, further confirming that TemMed-Bench remains substantially challenging and leaves ample headroom for future LVLMs.
>
> **Med-MIM [3]** provides a large instruction dataset and a benchmark for four types of multi-image abilities, and evaluates multiple LVLMs on multi-image QA. Its benchmark is still formulated as QA, whereas TemMed-Bench targets downstream temporal reasoning abilities that are closer to radiology practice, including generating change-focused follow-up reports and choosing longitudinal image pairs that match a given change description. In addition, we conduct a more comprehensive evaluation on our benchmark, including 12 LVLMs (6 proprietary and 6 open-source), whereas Med-MIM reports results for 6 LVLMs (1 proprietary and 5 open-source).
>
> **ICG-CXR [4]** defines a different task, Interpretable Counterfactual Generation (ICG), where models jointly generate counterfactual images that reflect a clinical hypothesis and corresponding interpretation texts that outline the visual changes induced by the hypothesis. Although ICG-CXR is also constructed from longitudinal medical studies, its task format and research motivation are substantially different from those of TemMed-Bench.
>
> We have revised the paper accordingly by softening the original wording, providing a more precise description, and adding the above discussion to the Related Work section to clarify how TemMed-Bench complements these concurrent efforts.

---

> ### Author Response · Authors · 2025-11-20
>
> **[Weakness 1.2] Concerns about Data Splitting**
>
>
> We thank the reviewer for raising this point. We would like to clarify that the split is performed at the instance level. As a result, cases from the same patient may appear in both the test set and the knowledge corpus.
>
> We fully understand the reviewer’s concern that such patient overlap could lead to near-duplicate visits and potentially inflate the RAG gains. At the same time, we note that in real-world clinical practice, it is often desirable to retrieve a patient’s other visits to help interpret the current condition. Our benchmark design intentionally follows this realistic medical RAG scenario.
>
> To directly address the concern about possible inflation, we conducted additional analyses and ablations:
>
>
>
> **(1) How often does same-patient retrieval occur?**
>
> In the VQA test set (2,000 cases), we find that only 185 cases (~9.2%) have a top-1 retrieved instance from the same patient as the target case.
>
> **(2) What happens if we forbid same-patient retrieval?**
>
> We re-evaluated two representative models, HealthGPT and GPT-4o, by excluding all same-patient instances from the retrieval pool. Below we report the results on the VQA task (accuracy, F1):
>
>
>
> | **Retrieval setting** | **HealthGPT** | | **GPT 4o** | |
> | :-: | :-: | :-: | :-: | :-: |
> | | **Text-only RAG** | **Multi-Modal RAG** | **Text-only RAG** | **Multi-Modal RAG** |
> | | **Acc/F1** | **Acc/F1** | **Acc/F1** | **Acc/F1** |
> | **Same-patient allowed** | 59.05/57.13 | 69.90/68.71 | 60.10/59.57 | 64.85/64.42 |
> | **Same-patient excluded** | 58.85/56.83 **↓0.2/0.3** | 69.25/67.99 **↓0.65/0.72** | 60.05/59.52 **↓0.05/0.05** | 64.60/64.21 **↓0.25/0.21** |
>
>
> The absolute changes of HealthGPT are around 0.3% in accuracy and 0.7% in F1. GPT-4o shows the same trend, with even smaller differences when instances from the same patient are excluded. We will include the full ablation table in the revised version.
>
> These results indicate that allowing same-patient data in the retrieval pool does not materially inflate the RAG gains reported in our benchmark. A key reason is that TemMed-Bench focuses on condition changes rather than single-visit pattern matching: even if the retrieved image pair comes from the same patient and the absolute conditions of these images are likely to be similar, the condition changes are not guaranteed to be similar. Therefore, based on these results and this analysis, the concern that retrieving instances from the same patients would inflate the RAG gains does not appear to be supported in our benchmark.
>
> We have added the above analysis and results to Appendix C.8 in the revision.

---

> > ### Author Response · Authors · 2025-11-20
> >
> > **[Weakness 2] Some Important Details**
> >
> >
> >
> > We thank the reviewer for pointing out these missing details. Our clarifications are as follows.
> >
> > **1. Choice of images:** For each selected report, we use the image pair from two consecutive examinations. Specifically, we take the most recent prior visit as the historical reference, since it is clinically the most relevant comparison for the current study.
> >
> > **2. Time gaps between examinations:** We appreciate the reviewer for raising this important point. We also considered analyzing performance as a function of the time interval between visits. However, we found that the exact date information in the CheXpert Plus dataset is inconsistent with the date_order metadata. We contacted the dataset author, Jean-Benoit Delbrouck, by email, and he confirmed that the exact dates were replaced with synthetic PHI in order to de-identify the data. Only the date order information is provided and has been validated. As a result, we are not able to reliably analyze performance across short vs. long intervals.
> >
> > **3. Image-pair selection task on existing LVLMs:** We do not modify the positional encoding of any LVLM. For Qwen2.5-VL and LLaVA-OneVision, the models natively support multiple image inputs. For Llama 3.2-Vision, although the official recommendation is to use a single image, the model can in fact process multiple images as input, and vLLM has added support for multiple-image inputs with Llama 3.2. For HealthGPT, HuatuoGPT-Vision, and LLaVA-Med, which all follow the LLaVA architecture, we apply the vision encoder to each image separately to extract features and then insert the resulting image tokens at specified positions among the text tokens. The detailed prompts for the image-pair selection task are provided in Appendix B.1.
> >
> > **4. Regular expression:** We thank the reviewer for raising this point. Our regular expressions are designed based on a broad empirical examination of the clinical language used in chest X-ray reports in the CheXpert dataset. In practice, we found that these reports are written in a highly standardized, template-like style, with a small and constrained set of adjectives used to describe temporal evolution (e.g., stable, unchanged, improved, worsened, resolved). This constrained reporting style makes a compact set of regular expressions an effective and reliable extraction approach in this specific setting.
> >
> > We adopt a regular expression-based approach primarily to guarantee that the extracted cases indeed contain the target condition changes, rather than to exhaustively capture all possible cases of that condition. In other words, our design prioritizes high precision over high recall. During development, we manually reviewed 100 reports extracted by our patterns and confirmed that all of them contained the intended target condition change (precision = 100%).
> >
> > To further enhance transparency and user confidence in our dataset, our released data will include both the extracted target condition change metadata and the corresponding original report text. In the revised version of the paper, we will also add a section with representative case studies, thereby making the behavior of the regex-based extraction more explicit.
> >
> >
> >
> > **5. Encoder for retrieval:** For the retrieval component, we adopt the retriever from MMed-RAG [1], which has been specifically fine-tuned for the radiology domain.
> >
> > [1] Xia P, et al. MMed-RAG: Versatile Multimodal RAG System for Medical Vision Language Models. International Conference on Learning Representations (ICLR), 2025.

---

> ### Comment · Reviewer_sLS7 · 2025-11-27
>
> I thank the authors for a comprehensive and empirically grounded rebuttal. They have revised the narratives regarding the novelty claims, addressed technical concerns regarding data leakage and implementation details, and also performed new ablation studies to empirically address the concern regarding performance inflation.
>
> I remain positive for accepting the paper.

---

> ### Author Response · Authors · 2025-11-27
>
> Thank you for your response! We sincerely appreciate your valuable feedback and your positive attitude toward our paper.

---

### Official Review · Reviewer_aQfR · 2025-10-28

**Soundness:** 2
**Presentation:** 3
**Contribution:** 2
**Rating:** 4
**Confidence:** 4

**Summary:**

This study proposes TemMed-Bench, the first benchmark specifically designed for evaluating Large Vision-Language Models' (LVLMs) ability to reason over temporal medical images by tracking changes in patients' conditions between different clinical visits. The benchmark consists of a test set across three distinct tasks:

1. Visual Question Answering (VQA): Challenges models to answer binary "yes" or "no" questions about condition changes between a historical and current image pair.
2. Report Generation: Requires models to generate a report analyzing condition changes observed between a historical and current image.
3. Image-pair Selection: A vision-centric task where models must select the image pair (from three options) that best matches a given medical statement describing a condition change.

They further includes a knowledge corpus of over 17,000 instances to support RAG. To enhance retrieval quality for temporal reasoning, the study introduces a novel "pairwise image retrieval method," which computes similarity based on both historical and current images to ensure that retrieved instances reflect similar condition changes, thereby enabling multi-modal retrieval augmentation. The evaluation reveals significant limitations in current LVLMs' temporal reasoning abilities and highlights the effectiveness of multi-modal RAG.

**Strengths:**

1. Large-scale Benchmark Construction: A total of 18,144 instances were constructed using an automated data collection pipeline applied to the CheXpert Plus dataset.

2. Multimodal RAG: The study introduces a multimodal RAG approach, incorporating a pairwise image retrieval method that considers similarity between historical and current images.

3. Temporal Reasoning Evaluation: Beyond single-visit image interpretation benchmarks, this work evaluates models’ ability to infer condition changes over time by comparing historical and current medical images.

**Weaknesses:**

This study makes a valuable contribution to evaluating temporal medical image reasoning in LVLMs. However, it exhibits several limitations:

1. Restricted Scope of "Temporal" Reasoning: The benchmark limits temporal analysis to a binary comparison between the current image and only the most recent prior visit. This simplified setting does not evaluate long-term progression, intermittent conditions, or multi-visit trends typical in clinical practice, reducing realism and the complexity of the reasoning task.

2. Lack of Robustness and Transparency in Data Collection: The dataset relies on handcrafted regular expressions to extract “condition change” sentences from medical reports, without any quantitative validation (precision, recall, F1) to assess extraction accuracy. Considering the variability of clinical language, this approach risks misinterpretation, extraction errors, and bias, ultimately harming dataset quality and representativeness. Clear documentation of the dataset extraction process is needed, along with transparent analysis of coverage and missed cases. Quantitative evidence demonstrating extraction performance would strengthen confidence in the dataset.

3. Insufficient Transparency and Expertise in Human Review: While the paper notes that human reviewers validated AI-generated VQA question-answer pairs, additional details about the review process would help support the reliability of this validation. Information such as the number of reviewers, their clinical expertise, review guidelines, and inter-rater agreement metrics (e.g., Cohen’s Kappa) would strengthen the methodological transparency.

**Questions:**

Were data ethics maintained in the proprietary LVLMs usage?

---

> ### Author Response · Authors · 2025-11-20
>
> We thank the reviewer for their insightful and constructive feedback. Our detailed responses to the questions and concerns are provided below.
>
> ---
>
>
> **[Weakness 1] Binary Comparison Concern**
>
> We thank the reviewer for this observation and agree that full longitudinal reasoning ideally involves more than two timepoints. In this work, we intentionally focus on the prevalent clinical scenario of comparing the current study with the most recent prior exam, and formulate all tasks around explicit change reasoning between these two visits. The poor performance of current LVLMs already in this setting suggests that robust two-timepoint temporal reasoning is a challenging and valuable first step. We will clarify in the revision that extending TemMed-Bench to multi-visit, sequence-level longitudinal trajectories with more than two time points is a key direction we are actively pursuing as future work.
>
>
> ---
>
>
> **[Weakness 2] Concern on Regular Expression-based Extraction**
>
>
> We thank the reviewer for raising the concern about the robustness and transparency of our regex-based extraction procedure.
>
> **(1) Variability of clinical language in chest X-ray reports**
>
> Our regular expressions are designed based on a broad empirical examination of the clinical language used in chest X-ray reports in the CheXpert dataset. **In practice, we found that these reports are written in a highly standardized, template-like style, with a small and constrained set of adjectives used to describe temporal evolution** (e.g., stable, unchanged, improved, worsened, resolved). Prior work has also documented that radiology reporting is often highly standardized and relies on a controlled lexicon rather than open-ended narrative descriptions[1][2][3]. This constrained reporting style makes a compact set of regular expressions an effective and reliable extraction approach in this specific setting.
>
> **(2) High-precision focus of the regular expression design**
>
> We adopt a regular expression-based approach primarily to guarantee that the extracted cases indeed contain the target condition changes, rather than to exhaustively capture all possible cases of that condition. In other words, our design prioritizes high precision over high recall. **During development, we manually reviewed around 100 reports extracted by our patterns and confirmed that all of them contained the intended target condition change (precision = 100%).**
>
> **To further enhance transparency and user confidence in our dataset, our released data will include both the extracted target condition change metadata and the corresponding original report text.** In the revised version of the paper, we have added a section in Appendix C.7 regarding this concern, thereby making the behavior of the regex-based extraction more explicit.
>
>
> [1] Bosmans JML, et al. Structured reporting: if, why, when, how—and at what expense? Insights into Imaging, 2012.
>
> [2] Yates A, et al. Structured reporting in portable chest radiographs: An essential tool in the diagnosis of COVID-19. European Journal of Radiology, 2021.
>
> [3] Johnson AEW, et al. MIMIC-CXR, a de-identified publicly available database of chest radiographs with free-text reports. Scientific Data, 2019.

---

> ### Author Response · Authors · 2025-11-20
>
> **[Weakness 3] Human Review Process Concern**
>
>
> We thank the reviewer for raising this important point and apologize for not providing a more detailed discussion of the faithfulness of the annotations in our benchmark. Below we clarify how our data construction pipeline is deliberately designed to minimize the need for additional expert re-labeling, while still maintaining high clinical fidelity.
>
> For the VQA task, we construct QA pairs by rephrasing each condition-change sentence in the target reports into a binary question whose answer is "yes" or "no". Concretely, we prompt an LLM to perform a purely syntactic transformation from the original sentence to a question, keeping the underlying clinical statement unchanged. For example, the sentence "Stable pulmonary edema" is converted into "Is the pulmonary edema in the patient stable?" with answer "Yes". This operation does not require any medical knowledge beyond preserving the meaning of the original sentence.
>
> Thanks to this data construction pipeline, human review of the data is quite simple and requires only very basic medical knowledge, such as familiarity with a limited number of diseases in the dataset. The human reviewer only needs to ensure that each question indeed targets the described change in condition and that the answer is consistent with the corresponding sentence in the original report. This human review process for the 2,000 VQA examples was carried out over a two-week period by a single person who had prior knowledge of the diseases covered in the dataset.
>
>
> We agree that an independent expert study would further strengthen the benchmark, and we will explicitly acknowledge the absence of such a dedicated expert validation as a limitation in the revision. At the same time, we will clarify that:
>
> - All labels are grounded in radiology reports written in clinical practice, rather than synthetic descriptions.
> - Our transformations are designed to be semantics-preserving and are accompanied by systematic manual quality control to ensure that all QA pairs and image-pair labels remain faithful to the original radiology reports.
>
> We have added a section in Appendix C.6 to clearly discuss the faithfulness and validation of the annotations in our benchmark.
>
>
>
> ---
>
>
>
> **[Question 1] Data Ethics Concern**
>
> Thank you for raising this point. The source data used in all our experiments come from the publicly released CheXpert Plus dataset, whose images and reports are fully de-identified by the dataset creators and contain no PII/PHI. We do not attach any additional patient-level identifiers or metadata when querying proprietary LVLM APIs, and the models are used only for inference on this de-identified benchmark, with no testing or training involving any identifiable clinical data.

---

> ### Author Response · Authors · 2025-11-27
>
> Dear Reviewer,
>
> We hope everything is going well for you. We are writing to gently follow up on our rebuttal and would be glad to provide any additional clarification if needed. Thank you very much for your time and effort.
>
> Authors

---

### Official Review · Reviewer_AJHW · 2025-10-30

**Soundness:** 3
**Presentation:** 3
**Contribution:** 3
**Rating:** 4
**Confidence:** 4

**Summary:**

This paper introduces TEMMED-BENCH, a benchmark to assess temporal reasoning by comparing historical vs. current medical images, demonstrates that multi-modal RAG substantially boosts performance, and includes manual QC for VQA, yet its single-modality/single-source design, limited failure-case analysis, and lack of broader SOTA baselines leave it vulnerable to score gaming and constrain generalizability.

**Strengths:**

1. Presents a benchmark that explicitly compares historical vs. current medical images to judge disease progression.
2. Systematically explores the impact of multi-modal RAG and demonstrates that it can markedly enhance task performance, providing a practical pathway for future agents/medical LLMs to handle such problems.
3. Includes manual quality control for the VQA subset, correcting approximately 10% of items and thereby improving the benchmark’s credibility.
4. Compared with datasets like U-Xray and MIMIC-CXR, this benchmark emphasizes reasoning grounded in retrieved evidence, making it less vulnerable to simple pattern-matching hacks.

**Weaknesses:**

1. The image modality is overly narrow, and the data source is single and easy to get; as a benchmark, it is easy to be reverse engineered and reward hacks, which may make it become obsolete quickly.
2. The paper mainly offers macro-level discussion of “why this is hard” and “which settings cause models to fail or become ineffective,” but lacks a systematic, category-wise failure analysis with concrete error cases. Since a benchmark is meant to drive LLM progress in a specific domain, the work should provide insights explaining why certain models fail.
3. Many evaluated LVLMs are not SoTA in 2025. For example, only Gemini 2.5 Flash was tested (not Gemini 2.5 Pro), and Claude 3.5 Sonnet (not 3.7 or later). Newer models should be included. Several open-source baselines are also relatively old, e.g., LLaVA-Med.

Minor:

2. The benchmark’s “temporal reasoning” setup is essentially **two-timepoint differencing**, rather than true longitudinal reasoning about disease progression over time.

**Questions:**

Please answer the following questions:

1. Because the dataset comes from a single, easily accessible source, the benchmark appears highly vulnerable to reverse engineering and score gaming. Do you have any safeguards to prevent this?

2. Please provide a failure-case analysis with concrete examples, and consider adding this section in future revisions.

3. If possible, include results from more advanced models to strengthen the evaluation.

---

> ### Author Response · Authors · 2025-11-20
>
> We appreciate the reviewer’s insightful comments and constructive feedback. Below, we respond to the questions and concerns raised.
>
> ---
>
> **[Weakness 1] [Question 1] Modality, data source and hacks**
>
>
> (1) Modality and data source.
>
> We thank the reviewer for pointing this out. We agree that the single-modality setting is a limitation and will state it clearly in the revised paper. Our primary goal, however, is to isolate temporal medical image reasoning, which is largely orthogonal to the choice of modality. Chest X-ray is one of the most common follow-up imaging exams, and CheXpert Plus provides large-scale longitudinal studies with reports that explicitly describe changes between visits, making it uniquely suitable for building a clean temporal benchmark at scale. Extending our modality-agnostic construction pipeline to CT, MRI, or ultrasound requires suitable longitudinal datasets and is therefore left as important future work, which we will highlight as a key extension direction in the revised version.
>
> (2) Reverse engineering and reward hacks.
>
> We thank the reviewer for raising this important concern. As discussed in Appendix C.1, we have already conducted an in-depth analysis of this issue. In contrast to many single-visit benchmarks that can be exploited via simple pattern matching or report retrieval, our benchmark explicitly emphasizes reasoning. As a result, models are encouraged to leverage retrieved information to reason over the images, rather than merely copying or lightly rephrasing it, which makes our benchmark more robust to such reward hacks.
>
> We also note that the underlying image source is indeed publicly accessible, and we do not rely on security through obscurity. Instead, the difficulty of the benchmark comes from the requirement to correctly understand subtle temporal changes between visits. This design choice makes overfitting to a fixed set of image-report pairs much less useful than in single-visit benchmarks.
>
>
> ---
>
>
> **[Weakness 2] [Question 2] Failure analysis**
>
> We appreciate the reviewer’s suggestion to include failure and error analyses. Due to space limitations at the rebuttal stage, we cannot provide detailed case studies here. However, we have included a failure analysis in Appendix C.9 with concrete error cases in the revised version, offering deeper insights into why certain models fail.
>
>
> ---
>
>
> **[Weakness 3] [Question 3] Results from More Advanced Models**
>
> We thank the reviewer for raising this point, and we fully understand the concern. To address this concern, we conducted additional experiments on three of the latest SOTA proprietary LVLMs available before the rebuttal deadline. The results are shown in the following tables.
>
>
> | | **VQA** | | **Report Generation** | | | | **Report Generation** |
> | :-: | :-: | :-: | :-: | :-: | :-: | :-: | :-: |
> | | **Acc** | **F1** |**BLEU** | **ROUGE-L** | **METEOR** | **Avg.** | **Acc** |
> | **Gemini 2.5 Pro** | **50.35** | **49.70** | 22.42 | 17.31 | 27.83 | **22.52** | **40.02** |
> | **Claude Sonnet 4.5** | **66.10** | **65.90** | 9.93 | 11.22 | 21.64 | **14.26** | **33.64** |
> | **GPT 5.1** | **59.35** | **59.35** | 16.03 | 14.41 | 28.51 | **19.65** | **41.58** |
>
>
> These results indicate that even the most advanced LVLMs are still struggling on TemMed-Bench and fully confirm that TemMed-Bench remains substantially challenging and leaves ample headroom for future LVLMs.
>
> We have added a section in Appendix C.5 to address this concern.
>
>
>
> ---
>
>
> **[Weakness 4] Two-Timepoint Concern**
>
>
> We thank the reviewer for this observation and agree that full longitudinal reasoning ideally involves more than two time points. However, we note that the two-time-point setting is already challenging enough for existing frontier LLMs, with the highest average BLEU/ROUGE-L/METEOR score for report generation being 20.67 and the best accuracy for image-pair selection being 39.33% in a three-option setting, both of which are well below an ideal level. This suggests that robust two-time-point temporal reasoning is a challenging and valuable first step.
>
> We also note that expanding the two-visit setting to three or more visits is trivial based on our existing data pipeline, which could be part of our future work. We have clarified in the revision that extending TemMed-Bench to multi-visit, sequence-level longitudinal trajectories with more than two time points is a key direction that we are actively pursuing as future work.

---

### Official Review · Reviewer_GNS5 · 2025-10-30

**Soundness:** 2
**Presentation:** 2
**Contribution:** 1
**Rating:** 2
**Confidence:** 4

**Summary:**

This paper presents TemMed-Bench, a new benchmark for evaluating temporal reasoning in medical vision-language models. Unlike previous datasets that test single-image understanding, TemMed-Bench requires models to compare image pairs from different patient visits to assess disease progression or improvement. It includes three task types: VQA, report generation, and image-pair selection. The authors evaluate open and proprietary LVLMs and find that most fail to perform reliable temporal reasoning. Incorporating multi-modal retrieval (image + text) improves performance over text-only retrieval but only modestly.

**Strengths:**

- **High practical relevance**: The proposed benchmark reflects real clinical workflows where patient history and temporal comparison are essential for diagnosis.
- **Comprehensive benchmark design**: It combines multiple task types (VQA, report generation, image-pair selection) to test LVLMs in medical reasoning from different perspectives.
- **Clear writing**: The paper is well written and clearly structured. The figures and tables are visually clear and effectively support the main points.

**Weaknesses:**

- **Benchmark difficulty**: The best-performing model, GPT-4o mini, already achieves around 80% accuracy (Table 3 and 4), suggesting that parts of the benchmark may not be very challenging for advanced models. In my opinion, this raises concerns about the benchmark’s long-term difficulty and its ability to differentiate performance among next-generation LVLMs.
- **No human expert validation**: The benchmark lacks human radiologist agreement or interpretability checks to confirm tasks and annotation correctness. Although the paper mentions a manual quality control for the VQA dataset, it focuses only on consistency checking rather than clinical verification by radiologists.
- **Limited task diversity beyond radiology**: The benchmark focuses mainly on chest X-ray data (CheXpert Plus), limiting the generalization to other modalities such as CT, MRI, or ultrasound.

**Questions:**

1. The paper shows that multi-modal RAG (combining image and text retrieval) improves performance over text-only retrieval. Could the authors clarify how the retrieved visual and textual evidence are integrated into the LVLM’s reasoning process?
2. How do the authors ensure that retrieval method does not introduce data leakage or spurious correlations from similar patient cases within the same dataset?
3. Given that GPT-4o mini already reaches around 80% accuracy, how likely is the benchmark to remain challenging and discriminative as future models improve?

---

> ### Author Response · Authors · 2025-11-20
>
> We thank the reviewer for their insightful comments and constructive feedback. Below, we address the questions and concerns raised by the reviewer.
>
> ---
>
> **[Weakness 1][Question 3] Benchmark difficulty**
>
> We fully understand the reviewer’s concern and would like to clarify that only one frontier proprietary model, GPT-4o mini, reaches 79.15% on the binary VQA task, whereas most models remain close to chance level (around 50–60%). TemMed-Bench is explicitly a three-task benchmark: VQA is designed as a relatively easier entry-level task to ensure broad model coverage, while the other two tasks are far from solved. The highest average BLEU/ROUGE-L/METEOR score for report generation is 20.67, and the best accuracy for image-pair selection is 39.33% in a three-option setting.
>
> At the same time, we acknowledge that this concern is valid. To directly address it, we conducted additional experiments on three of the latest SOTA proprietary LVLMs available before the rebuttal deadline. The results are shown in the following tables.
>
> | | **VQA** | | **Report Generation** | | | | **Report Generation** |
> | :-: | :-: | :-: | :-: | :-: | :-: | :-: | :-: |
> | | **Acc** | **F1** |**BLEU** | **ROUGE-L** | **METEOR** | **Avg.** | **Acc** |
> | **Gemini 2.5 Pro** | **50.35** | **49.70** | 22.42 | 17.31 | 27.83 | **22.52** | **40.02** |
> | **Claude Sonnet 4.5** | **66.10** | **65.90** | 9.93 | 11.22 | 21.64 | **14.26** | **33.64** |
> | **GPT 5.1** | **59.35** | **59.35** | 16.03 | 14.41 | 28.51 | **19.65** | **41.58** |
>
> These results indicate that even the most advanced LVLMs are still struggling on TemMed-Bench and fully confirm that TemMed-Bench remains substantially challenging and leaves ample headroom for future LVLMs.
>
> We have added a section in Appendix C.5 to address this concern.

---

> ### Author Response · Authors · 2025-11-20
>
> **[Weakness 2] No human expert validation**
>
> We thank the reviewer for raising this important point and apologize for not providing a more detailed discussion of the faithfulness of the annotations in our benchmark. **Below we clarify how our data construction pipeline is deliberately designed to minimize the need for additional expert re-labeling, while still maintaining high clinical fidelity.**
>
> **(1) Report generation task.**
>
> **For the report generation task, we directly select radiology reports from the CheXpert Plus dataset in which every sentence is a condition change description sentence, without generating new clinical findings.** These reports are authored and validated in routine clinical workflow by radiologists, and our pipeline only selects those in which every sentence describes a change in the patient’s condition across visits, without modifying their content. Thus, the correctness of the target reports is inherited from the original dataset, and no extra expert re-annotation is introduced by our benchmark.
>
> **(2) VQA task.**
>
> For the VQA task, we construct QA pairs by rephrasing each condition-change sentence in the target reports into a binary question whose answer is "yes" or "no". **Concretely, we prompt an LLM to perform a purely syntactic transformation from the original sentence to a question, keeping the underlying clinical statement unchanged.** For example, the sentence "Stable pulmonary edema" is converted into "Is the pulmonary edema in the patient stable?" with answer "Yes". **This operation does not require any medical knowledge beyond preserving the meaning of the original sentence.** Thanks to this data construction pipeline, human review of the data is quite simple and requires only very basic medical knowledge, such as familiarity with a limited number of diseases in the dataset. The human reviewer only needs to ensure that each question indeed targets the described change in condition and that the answer is consistent with the corresponding sentence in the original report. This human review process for the 2,000 VQA examples was carried out over a two-week period by a single person who had prior knowledge of the diseases covered in the dataset.
>
>
> **(3) Image-pair selection task.**
>
> **For the image-pair selection task, we deliberately avoid using LLMs to interpret reports. Instead, we design regular-expression patterns over a fixed set of change-keywords and pathology terms, tailored to the highly standardized wording of radiology reports.** During development, we manually reviewed around 100 reports extracted by our patterns and confirmed that all of them contained the intended target condition change (precision = 100%). This pattern-based construction gives us a deterministic mapping from the report text to the label, avoiding the additional uncertainty that LLM-based interpretation would introduce and thereby reducing the need for radiologist re-verification.
>
>
>
>
> We agree that an independent expert study would further strengthen the benchmark, and we will explicitly acknowledge the absence of such a dedicated expert validation as a limitation in the revision. At the same time, we will clarify that:
>
> - **All labels are grounded in radiology reports written in clinical practice, rather than synthetic descriptions.**
> - **Our transformations are designed to be semantics-preserving and are accompanied by systematic manual quality control to ensure that all QA pairs and image-pair labels remain faithful to the original radiology reports.**
>
> We have added a section in Appendix C.6 to clearly discuss the faithfulness and validation of the annotations in our benchmark.

---

> ### Author Response · Authors · 2025-11-20
>
> **[Weakness 3] Limited task diversity beyond radiology**
>
> We thank the reviewer for pointing this out. We agree that the single-modality setting is a limitation and will state it clearly in the revised paper. Our primary goal, however, is to isolate temporal medical image reasoning, which is largely orthogonal to the choice of modality. Chest X-ray is one of the most common follow-up imaging exams, and CheXpert Plus provides large-scale longitudinal studies with reports that explicitly describe changes between visits, making it uniquely suitable for building a clean temporal benchmark at scale. Extending our benchmark to CT, MRI, or ultrasound requires suitable longitudinal datasets and is therefore left as important future work, which we have highlighted as a key extension direction in the revised version.
>
> ---
>
> **[Question 1] Clarify how the retrieved visual and textual evidence are integrated into the LVLM’s reasoning process**
>
> In our multi-modal RAG setting, the LVLM receives a single sequence that includes the query image pair and question, as well as the top-k retrieved cases, each containing an image pair and its report. This allows the model to visually compare the query pair against retrieved pairs that exhibit similar or different patterns of condition change, rather than only reading their textual descriptions. Through cross-modal attention over the query images, retrieved images, and reports, the LVLM can better judge which retrieved cases are truly relevant to the current visual findings and which should be down-weighted. In contrast, text-only RAG can only rely on the retrieved reports without this visual grounding, which explains why multi-modal RAG leads to larger gains in our experiments.
>
>
>
> ---
>
>
>
>
> **[Question 2] Data leakage concern**
>
> We thank the reviewer for raising this concern. We respond to this concern from three perspectives.
>
> **(1) No instance-level leakage.** We construct 18,144 temporal instances from CheXpert Plus and hold out 1,000 instances as the test set, using only the remaining 17,144 instances to build the retrieval pool. Test instances, including their image pairs, associated reports, and derived questions, are never inserted into the retrieval pool or used to train the retriever, so the exact test pair and its report cannot be retrieved.
>
> **(2) Labels are defined on temporal changes, not single images.** Each label describes the change between two visits, rather than the status of an isolated image. Even if the retriever finds a visually similar patient, the retrieved report corresponds to the change pattern of a different pair of visits, which generally does not determine the ground-truth change for the test pair. The model must still reason over the historical-current image pair to answer the question.
>
> **(3) Similar patients do not imply label copying.** Our retrieval uses only image content, without using patient IDs or other metadata. Thus, while similar or even same-patient cases may exist across splits, as in realistic clinical RAG scenarios, this does not constitute label leakage, because the exact test instance is unseen and its change label cannot be obtained by simple copying.
>
> We have added a section in Appendix C.8 of the revised manuscript to address this concern.

---

> ### Author Response · Authors · 2025-11-27
>
> Dear Reviewer,
>
> We hope everything is going well for you. We are writing to gently follow up on our rebuttal and would be glad to provide any additional clarification if needed. Thank you very much for your time and effort.
>
> Authors

---

### Author Response · Authors · 2025-11-21
**Global Response**

### Updated manuscript

Based on the reviewers’ helpful and detailed comments, we have revised our manuscript and uploaded an updated version. **All modifications are highlighted in blue. In particular, we have adjusted several parts of the main text and added Appendices C.5, C.6, C.7, C.8, C.9, and C.10.** We hope these changes enhance the clarity and readability of the paper and adequately address the reviewers’ questions and concerns.

We sincerely appreciate the reviewers’ time and constructive feedback, which have significantly improved the presentation of our work. We remain happy to engage in further discussion to clarify any remaining issues regarding our submission.

---

### Author Response · Authors · 2025-12-01
**A Summary of Review, Rebuttal, and Discussion**

Dear Area Chair,

We are deeply regretful about the recent information leakage within the community. To support a fair and thorough evaluation, we would like to summarize the reviews and our rebuttal.

Reviewers commended the paper for the **clinical relevance and realism of its setting (GNS5, AJHW, sLS7)**, noting that the benchmark closely mirrors longitudinal radiology workflows where temporal comparison of historical and current images is central, rather than single-visit reasoning. They highlighted the **comprehensive and large-scale benchmark design (GNS5, aQfR, sLS7)**, covering VQA, report generation, and a novel, vision-centric image-pair selection task, with 18,144 instances in total. Reviewers further emphasized the focus on **temporal and evidence-grounded reasoning (AJHW, aQfR)**, including explicit assessment of disease progression, pairwise retrieval of longitudinal studies, and evaluation protocols that prioritize reasoning over superficial pattern matching. They also valued the **systematic study of multimodal RAG (AJHW, aQfR)**, showing that integrating retrieved images and reports can substantially boost performance and provide a practical pathway for future medical LLMs/agents. Finally, reviewers appreciated the **careful benchmark curation (AJHW) and clear presentation (GNS5, sLS7)**, including manual quality control, concrete task definitions, corpus statistics, straightforward evaluation protocols, and overall clarity of writing, figures, and tables.

During rebuttal, we have carefully addressed the main concerns raised by the reviewers, including **(1) the faithfulness and validity of our annotations, (2) the long-term difficulty of TemMed-Bench, and (3) our data splitting strategy and potential data leakage.** Our extended analysis was positively received by reviewers sLS7 and AJHW.

- **Reviewer sLS7** recognized the empirical results and ablation studies presented during the rebuttal and remains positive about accepting the paper.

- **Reviewer AJHW** raised their score **from 4 to 6 on 20 Nov 2025** at 22:52 EST, as recorded in the revision history. Since the reviewer did not mention this in a comment, we would like to bring it to your attention and would sincerely appreciate it if the AC could take this into account. Given that this update took place approximately one week prior to the leak incident, we believe it is trustworthy and reflects the strengths of our work.

Thank you again for your time and for your service to the community. Please let us know if any additional clarification is needed.

Best regards,

Authors

---

### Meta-Review · Area_Chair_cq76 · 2026-01-07

**Summary:**

This paper introduces a temporal medical image reasoning benchmark with three tasks: VQA, report generation, and image-pair selection. The reviewers acknowledge the practical importance of such a temporal change reasoning benchmark and the comprehensive evaluation. The major concerns are mainly around limited modality or data source, the difficulty of the benchmark, the lack of human validation, the lack of SoTA baselines, etc. The rebuttal addressed most of the concerns.

There are still a few remaining major issues. 1) The authors missed a few relevant works, particularly MMXU, on medical temporal reasoning and wrongly claimed their work is the first one. Although they tuned down the claim to be the first multi-task benchmark, it makes their first contribution weak. It seems the paper needs a major rewrite to articulate its contributions against MMXU and emphasize the major contributions on the new tasks. What are the differences for the VQA task? Can the two additional tasks be built on top of MMXU? 2) Although the authors explained the quality of regular expression and emphasized their 100 reviewed reports are of 100% precision, to ensure the quality of the benchmark, it is better to do a few rounds of validations, instead of only based on one round of a small batch of 100 reports for human validation. Also, the test set needs to be 100% verified by humans.

Overall, although this work has high potential, unfortunately, the paper is not ready in its current shape, given the remaining major concerns.

**Reviewer Concerns:**

See above.

**Reviewer Scores:**

The first two reviewers might consider upgrading their scores slightly.

---

### Decision · Program_Chairs · 2026-01-26

Reject